# ATP-free in vitro biotransformation of starch-derived maltodextrin into poly-3-hydroxybutyrate via acetyl-CoA

Xinlei Wei [1,2], Xue Yang[2], Congcong Hu[1,2,3], Qiangzi Li[1,2,4], Qianqian Liu[2], Yue Wu[2], Leipeng Xie[1,2], Xiao Ning[1,2,4], Fei Li[1,2], Tao Cai [2], Zhiguang Zhu [1,2,4,5], Yi-Heng P. Job Zhang[1,2,4,5], Yanfei Zhang [2,4,5], Xuejun Chen[2] & Chun You [1,2,4,5] ✉

In vitro biotransformation (ivBT) facilitated by in vitro synthetic enzymatic biosystems (ivSEBs) has emerged as a highly promising biosynthetic platform. Several ivSEBs have been constructed to produce poly-3-hydroxybutyrate (PHB) via acetyl-coenzyme A (acetyl-CoA). However, some systems are hindered by their reliance on costly ATP, limiting their practicality. This study presents the design of an ATP-free ivSEB for one-pot PHB biosynthesis via acetyl-CoA utilizing starch-derived maltodextrin as the sole substrate. Stoichiometric analysis indicates this ivSEB can self-maintain $NADP^+$/NADPH balance and achieve a theoretical molar yield of 133.3%. Leveraging simple one-pot reactions, our ivSEBs achieved a near-theoretical molar yield of 125.5%, the highest PHB titer (208.3 mM, approximately 17.9 g/L) and the fastest PHB production rate (9.4 mM/h, approximately 0.8 g/L/h) among all the reported ivSEBs to date, and demonstrated easy scalability. This study unveils the promising potential of ivBT for the industrial-scale production of PHB and other acetyl-CoA-derived chemicals from starch.

In vitro biotransformation (ivBT) mediated by in vitro synthetic enzymatic biosystems (ivSEBs) refers to the utilization of multiple enzymes for the production of biocommodities along the designed artificial metabolic pathways[1,2]. The reaction pathways for ivBTs are typically designed based on pre-existing metabolic pathways with necessary modifications[3], and can be attained by combining multiple reaction modules, each comprising a set of enzymes for a certain catalytic purpose[4]. Unshackled from cell viability and complexity, ivBT offers a high degree of engineering flexibility[5,6], facile control and optimization[7,8], and excellent tolerance to cell-toxic products[9,10]. Moreover, with a reasonably designed pathway and well-optimized

reaction conditions, ivBT can achieve fast reaction rates[11,12] and high product yields[13,14]. These distinctive attributes endorse ivBT as a promising state-of-the-art platform for biosynthesis[2].

Acetyl-coenzyme A (acetyl-CoA) is an important platform chemical that can be employed to synthesize a variety of chemicals, such as poly-3-hydroxybutyrate (PHB), n-butanol, lipids, and isoprenoids[15,16]. PHB, a natural biodegradable and biocompatible polyester with extensive potential applications as a feed additive[17,18], a bone implant material[19,20], an antimicrobial agent[21], and a drug delivery matrix[22], has been synthesized via acetyl-CoA by using several ivSEBs. Most of these ivSEBs share the same downstream cascade pathway consisting of

[1]In vitro Synthetic Biology Center, Tianjin Institute of Industrial Biotechnology, Chinese Academy of Sciences, 32 West 7th Avenue, Tianjin Airport Economic Area, Tianjin 300308, People's Republic of China. [2]Tianjin Institute of Industrial Biotechnology, Chinese Academy of Sciences, 32 West 7th Avenue, Tianjin Airport Economic Area, Tianjin 300308, People's Republic of China. [3]Key Laboratory of Industrial Fermentation Microbiology, Ministry of Education, Tianjin Industrial Microbiology Key Laboratory, College of Biotechnology, Tianjin University of Science and Technology, Tianjin 300457, People's Republic of China. [4]University of Chinese Academy of Sciences, 19A Yuquan Road, Shijingshan District, Beijing 100049, People's Republic of China. [5]National Technology Innovation Center of Synthetic Biology, Tianjin 300308, People's Republic of China. ✉e-mail: you_c@tib.cas.cn

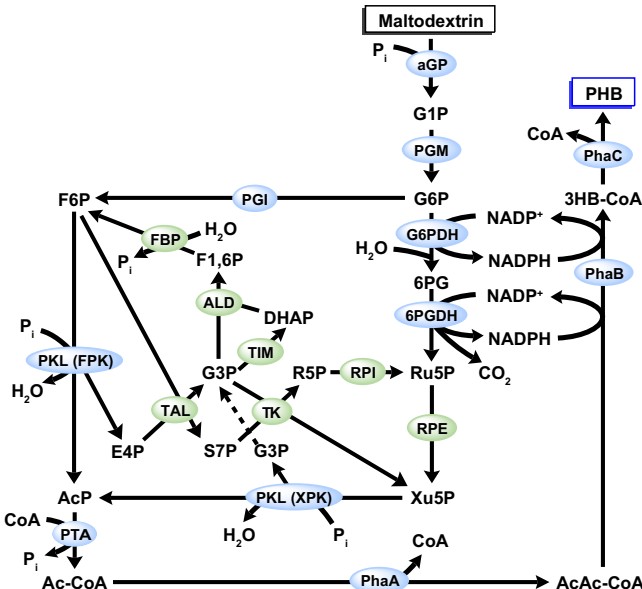

**Fig. 1 | Schematic of the in vitro synthetic enzymatic pathway for the production of PHB from maltodextrin.** Key enzymes for PHB production are marked in light blue. Enzymes for the carbon rearrangement are marked in green. Abbreviations of enzymes are provided in Supplementary Table 1. $P_i$ inorganic phosphate, G1P glucose 1-phosphate, G6P glucose 6-phosphate, 6PG 6-phosphogluconate, Ru5P ribulose 5-phosphate, Xu5P xylulose 5-phosphate, F6P fructose 6-phosphate, AcP acetyl phosphate, F1,6P fructose 1,6-bisphosphate, DHAP dihydroxyacetone phosphate, G3P glyceraldehyde 3-phosphate, R5P ribose 5-phosphate, E4P erythrose 4-phosphate, S7P sedoheptulose 7-phosphate, CoA coenzyme A, Ac-CoA acetyl-CoA, AcAc-CoA acetoacetyl-CoA, 3HB-CoA 3-hydroxybutyryl-CoA.

acetyl-CoA acetyltransferase (PhaA), acetoacetyl-CoA reductase (PhaB), and PHB synthase (PhaC) to catalyze the NADPH-dependent conversion of acetyl-CoA into PHB. Different substrates and pathways have been proposed upstream of this reaction cascade for acetyl-CoA synthesis. For example, Satoh et al. [23] used a single enzyme, acetyl-CoA synthase, for the ATP-dependent conversion of acetate and coenzyme A (CoA) into acetyl-CoA. In their method, ATP was directly added as a substrate, and NADPH required by the PhaB reaction was regenerated from glucose using glucose dehydrogenase (GDH). Based on this system, Li et al. [24] added natural thylakoid membranes (TMs) and polyphosphate:AMP phosphotransferase for the simultaneous light-driven regeneration of NADPH and ATP. However, this ivSEB might suffer from substrate inhibition[24], tedious preparation and instability of TMs, and reduced light penetration efficiency during scale-up[25]. Zhang et al. designed a chemical-biological hybrid pathway to produce PHB from carbon dioxide ($CO_2$), in which an ivSEB was used to convert the $CO_2$-derived methanol to PHB[26]. In this self-driven ivSEB, methanol was converted to both PHB and formic acid, and the latter was used for NADPH regeneration. Due to the toxicity of methanol, the reaction must be performed in a two-pot, three-step mode, indicating the difficulty in scale-up[26]. Opgenorth et al. [27] proposed another self-driven ivSEB using pyruvate as the sole substrate to provide both acetyl-CoA and the reducing power by pyruvate dehydrogenase (PDH) complex. This ivSEB contained a specifically designed purge valve module to maintain the redox balance. Because of the difficulty of preparing the PDH complex[10] and the relatively high cost of pyruvate, this system was later surpassed by a more delicate ivSEB developed by the same research group[28]. In this ivSEB containing 19 enzymes, glucose was activated by glucose kinase in the presence of ATP into glucose 6-phosphate (G6P), which was then used for the production of acetyl-CoA as well as the regeneration of ATP and NADPH. It was reported that the theoretical yields of PHB from glucose is 100% and 66.7% in

terms of molarity and carbon atoms, respectively. Among all previously documented ivSEBs, this glucose-based system demonstrated the highest PHB titer of 93.8 monomer equivalent, accompanied by a notable molar yield of around 90%. Nevertheless, this system might suffer from instability and expense of ATP, and was complicated by the addition of two coenzyme purge valves for NADP(H) regulation, thereby posing difficulties in industrialization.

In this study, we designed and constructed an ATP-free ivSEB containing 17 enzymes for the biosynthesis of PHB from maltodextrin (a derivative of starch) via acetyl-CoA. Computational analysis of stoichiometry demonstrated that our ivSEB is capable of self-maintaining the redox balance, obviating the need for coenzyme regulation modules. Thus, this designed ivSEB enabled theoretical yields of PHB with values of 133.3% and 88.9% in terms of molarity and carbon atoms, respectively. Upon optimization facilitated by a fitted kinetic model, our ivSEB produced 74.9 mM (6.4 g/L) PHB at a production rate of 9.4 mM/h (0.8 g/L/h), with a near-theoretical molar yield of 125.5% based on maltodextrin consumption. After the addition of two auxiliary enzymes for the complete utilization of substrate, 118.8 mM (10.2 g/L) PHB was produced through the one-pot reaction with a molar yield of 120.1%. The PHB titer was further increased to 208.3 mM (17.9 g/L) when doubling the substrate concentration, indicating easy scale-up and promising industrial potential. To the best of our knowledge, our system achieved the fastest reaction rate and the highest PHB titer among all the ivSEBs for PHB production reported to date. Our findings have the potential to advance the establishment of a highly efficient ivBT platform for the biosynthesis of acetyl-CoA-derived products on industrial scale.

## Results
### Pathway design for the in vitro PHB production from maltodextrin

The pathway utilized by our designed ivSEB for the ATP-free production of PHB from maltodextrin is shown in Fig. 1. Enzymatic reactions of this pathway are described as follows: (1) maltodextrin is phosphorylated to glucose 1-phosphate (G1P) by α-glucan phosphorylase (αGP) in the presence of inorganic phosphate ($P_i$), followed by the conversion of G1P to G6P catalyzed by phosphoglucomutase (PGM); (2) glucose 6-phosphate dehydrogenase (G6PDH), 6-phosphogluconate dehydrogenase (6PGDH), and ribose 5-phosphate 3-epimerase (RPE), which are part of the natural pentose phosphate pathway (PPP), convert a portion of G6P to xylulose 5-phosphate (Xu5P), accompanying with the generation of NADPH and $CO_2$; (3) the other portion of G6P is isomerized to fructose 6-phosphate (F6P) by phosphoglucose isomerase (PGI); (4) both Xu5P and F6P can be used by the bifunctional phosphoketolase (PKL) to generate acetyl-phosphate (AcP), releasing erythrose 4-phosphate (E4P) and glyceraldehyde 3-phosphate (G3P), respectively; (5) to completely use the glucose units in maltodextrin, E4P and G3P are respectively recycled to F6P and Xu5P by carbon rearranging enzymes of the natural PPP, which are transaldolase (TAL), transketolase (TK), ribose 5-phosphate isomerase (RPI), triose phosphate isomerase (TIM), fructose-bisphophate aldolase (ALD), fructose 1,6-bisphosphatase (FBP), and RPE; (6) AcP produced by PKL is then converted to acetyl-CoA by phosphate acetyltransferase (PTA); (7) finally, the PhaA-PhaB-PhaC cascade uses acetyl-CoA and NADPH for the production of PHB. The whole process requires 17 enzymes (Supplementary Table 1), and has an intricate pathway network in which many intermediates can be utilized by multiple enzymes (Fig. 1). For instance, G6P is used by both G6PDH and PGI, while F6P is used by both PKL and TAL. Analysis of the Gibbs energy change ($\Delta_r G^\circ$) and equilibrium constant ($k_{eq}$) suggests that most reactions in our designed ivSEB pathway are thermodynamically favorable (Supplementary Table 2). However, the precise stoichiometric profiles of the substrate, coenzymes, intermediates, and products in this pathway were unclear.

## Stoichiometric analysis of the designed pathway

Before the experimental validation of our ivSEB, the stoichiometric profiles of substrate, coenzymes, intermediates, and products were analyzed to enhance our understanding of this system. Due to the intricacy of the designed pathway, stoichiometric analysis was facilitated by a computational model constructed by COPASI[29]. As the stoichiometric properties of a system are independent of the reaction kinetics involved[30], accurate enzymatic kinetic parameters are not required to reflect the real-world situations. Therefore, we simplified the model by using basic kinetic functions (Supplementary Table 3) and randomly assigning kinetic parameter values (see Methods). Because PKL is a bifunctional enzyme, the constructed model suggested reaction stoichiometries of two extreme situations: (1) PKL functions as a Xu5P-specific PKL (XPK); (2) PKL functions as a F6P-specific PKL (FPK) (Supplementary Table 4). Aside from the stoichiometric coefficients of XPK and FPK, the only difference between the XPK and PFK pathways pertains to the stoichiometric coefficient of the bifunctional TK. In the XPK pathway, TK-2 reaction (defined as Xu5P + E4P = G3P + F6P) has a negative coefficient of −2, indicating the net reaction was in the reverse direction, producing 2 Xu5Ps and 2 E4Ps from 2 G3Ps and 2 F6Ps. In the FPK pathway, the positive coefficient of TK-2 suggests the net reaction is towards the production of G3P and F6P. Consequently, we were able to derive the precise stoichiometric profiles of all chemicals in our designed pathway (Fig. 2), providing information on the theoretical distributions of reaction intermediates at pathway branch points. In both the XPK (Fig. 2a) and the FPK (Fig. 2b) pathways, the stoichiometric coefficients of αGP-, PGM-, PGI-, and G6PDH-catalyzed reactions are 3, 3, 1, 2, respectively, implying that for every 3 G6Ps produced by the αGP-PGM cascade, one is used by PGI and the other two are used by G6PDH. A stoichiometric coefficient of 8 for the XPK reaction in the XPK pathway (Fig. 2a) suggests for every 3 glucose equivalents of maltodextrin consumed by αGP, there will be 8 Xu5Ps available for the XPK reaction, generating 8 AcPs. Among these 8 AcPs, 2 AcPs are from the G6PDH-6PGDH-RPE-XPK cascade, while the other 6 AcPs are produced from 1 F6P and 2 G3Ps by XPK and a carbon rearrangement module consisting of 7 enzymes (TAL, TK, RPI, RPE, TIM, ALD, and FBP) as shown in Fig. 2c. Similarly, the stoichiometric coefficient of 8 for the FPK reaction in the FPK pathway (Fig. 2b) suggests 8 AcPs are produced from 8 F6Ps. One of these F6Ps is from the PGM-PGI cascade, while the other 7 F6Ps are produced from 2 Ru5Ps and 8 E4Ps by the carbon rearrangement module containing the same enzymes as that in the XPK pathway but reacted in a different pattern (Fig. 2d). The resulting 8 AcPs are subsequently converted to 8 acetyl-CoAs, and then used to synthesize 4 monomer equivalents of PHB through the PhaA-PhaB-PhaC cascade. In both the XPK and the FPK pathways, NADP$^+$/NADPH, CoA, and P$_i$ are all stoichiometrically balanced (Fig. 2). Therefore, the overall stoichiometric equation of our designed pathway can be written as $(C_6H_{10}O_5)_n = (C_6H_{10}O_5)_{n-3} + 4 (C_4H_6O_2) + 2 CO_2 + 3 H_2O$, in which $C_6H_{10}O_5$ is the glucose unit of maltodextrin, and $C_4H_6O_2$ is the monomer unit of PHB. This equation suggests our designed ivSEB ideally consumes every 3 glucose equivalents of maltodextrin for the generation of 4 monomer equivalents of PHB, thus the theoretical yield of PHB from maltodextrin is 133.3% and 88.9% in terms of molarity and carbon atoms, respectively.

## Proof-of-concept production of PHB from maltodextrin

Next, a proof-of-concept experiment was conducted to test our designed ivSEB. The reaction was carried out in one pot with around 55.6 mM (in terms of glucose equivalent, hereinafter the same) of isoamylase (IA)-debranched maltodextrin as substrate. Purified enzymes (Supplementary Fig. 1) were used for the experiments. Each enzyme was loaded at a final concentration of 1 U/mL, except for PhaA, which was loaded at 1 mg/mL (0.78 mU/mL) due to its low specific activity. Because PHB is water-insoluble[31], the absorbance of reaction

mixture at 600 nm (OD$_{600}$) was used to roughly estimate the amount of PHB produced[28].

As illustrated in Fig. 2, some enzymes in our ivSEB are not essential for PHB production but can boost the product yield. Based on the stoichiometric profiles of the intact system, we further inferred that: (1) for the XPK pathway with only the essential enzymes (i.e. without TK, TIM, ALD, FBP, TAL, RPI, and PGI; referred to as XPK essential), the consumption of every 2 glucose equivalents of maltodextrin results in the production of 1 monomer equivalent of PHB, corresponding to a molar yield of 50% (Fig. 3a); (2) for the FPK pathway with only the essential enzymes (i.e. without TK, TIM, ALD, FBP, TAL, RPI, and RPE; denoted as FPK essential), a theoretical molar yield of 40% is expected (Fig. 3b). These predictions were supported by the proof-of-concept experimental results. Specifically, the XPK essential system exhibited slightly higher OD$_{600}$ values than the FPK essential system, while the intact system displayed the highest OD$_{600}$ readings (Fig. 3c). These results indicated the feasibility of producing PHB from maltodextrin and the reliability of the stoichiometric analysis of our designed pathway.

## Quantitative evaluation of PHB

Subsequently, quantitative investigations were carried out using 100 mM IA-debranched maltodextrin as the substrate. PHB in the sample was methanolyzed by heating at 95 °C in the presence of methanol and sulfuric acid, and detected by gas chromatography, and its amount was compared with that of commercial PHB standards (Supplementary Fig. 2) for quantification. Using the same enzyme concentrations as those in the proof-of-concept experiment, PHB titer reached a maximal value of 23.0 ± 0.1 mM (in terms of monomer equivalent, hereinafter the same) at 8 h when 21.7 ± 2.2 mM maltodextrin was consumed (Supplementary Fig. 3a, b). Despite that only a small fraction of maltodextrin was consumed, the molar yield of PHB based on the consumed maltodextrin was 106.0%. To enhance the PHB titer, the loading amounts of all enzymes were increased by five-fold. In this case, the production of PHB almost finished at around 4 h when 53.6 ± 2.1 mM maltodextrin was consumed and 62.8 ± 3.3 mM PHB was generated (Supplementary Fig. 3c, d, e), representing a molar yield of 117.2%. Based on our previous study that an ivSEB with the same αGP could consume up to roughly 60% of IA-treated maltodextrin[32], it seemed that this ivSEB accomplished near-theoretical substrate consumption when enzyme concentrations were increased by five-fold.

## Effect of cofactor input on the production of PHB

In the course of the aforementioned quantitative PHB production experiments, the reaction solutions were initially supplemented with several cofactors at the onset (0 h), including 2 mM NADP$^+$, 0.5 mM CoA, and 0.5 mM TPP. This supplementation strategy was adopted based on our hypothesis that the balance of these cofactors, especially NADP$^+$/NADPH, could be self-sustained within our ivSEB. Consequently, we investigated the impact of varying concentrations of NADP$^+$ on both the initial reaction rate and the ultimate yield of PHB. At five-fold enzyme concentrations, adjustment of NADP$^+$ input concentrations within the range of 0 – 20 mM resulted in varied initial reaction rates (Supplementary Fig. 4a) but similar PHB final titers (Supplementary Fig. 4b). Excessive NADP$^+$ was found to slow down the initial reaction rate, possibly due to its inhibitory effect on PhaB[23]. To our surprise, PHB could be produced at a high titer even without external supplementation of NADP$^+$, despite a relatively slow rate. This observation may be attributed to the presence of trace amount of NADP(H) in the purified G6PDH and 6PGDH, which could not be removed during purification process, including dialysis. These results suggested that our ivSEB has the capability to maintain a redox balance without additional regulation modules, such as a purge valve.

The influence of the other two cofactors, CoA and thiamine pyrophosphate (TPP), on PHB production was then explored by varying

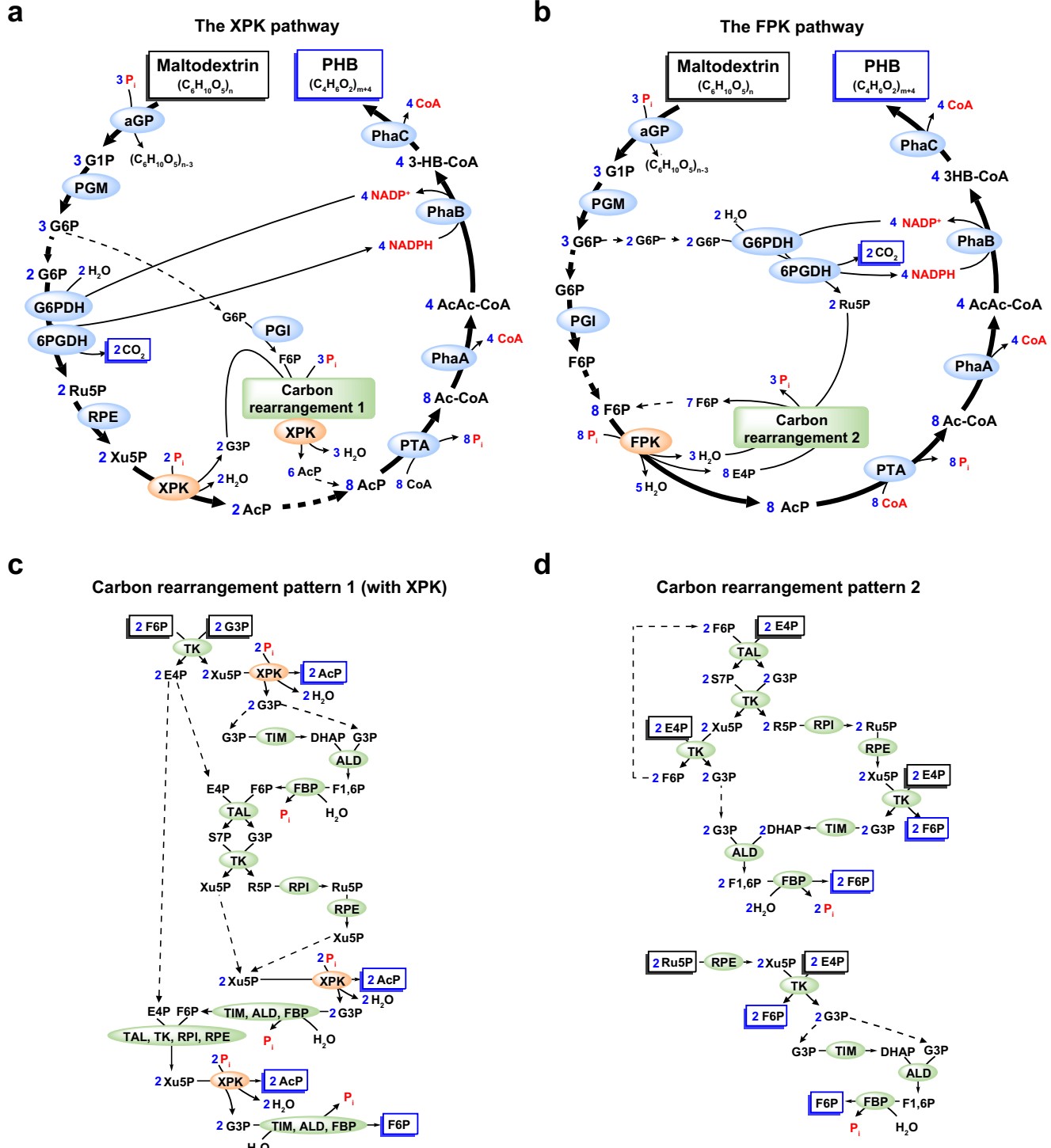

**Fig. 2 | Schematic of the in vitro synthetic enzymatic pathways for the stoichiometric production of PHB from maltodextrin.** Stoichiometric coefficients of enzymatic reactions are shown in blue. NAPD(H), CoA, and $P_i$ are highlighted in red to visibly display their cycles in the reaction pathways. Carbon-containing substrates and products are marked in rectangles. **a** Schematic of the XPK pathway, in which PKL is specifically active towards Xu5P. **b** Schematic of the FPK pathway, in which PKL is specifically active towards F6P. **c** Schematic of the pathway for the synthesis of AcP from F6P and G3P by carbon rearrangement pattern 1 and XPK. Cascade reactions that repeatedly occurred (for example, the TIM-ALD-FBP reaction cascade) were drawn in a simplified mode. **d** Schematic of the pathway for the conversion of E4P and Ru5P to F6P by carbon rearrangement pattern 2.

their concentrations. Within the range of 0 – 2 mM, the supplementation of 0.5 mM CoA at the onset of the reaction facilitated the attainment of both the highest initial rate of PHB production (Supplementary Fig. 5a) and the maximum PHB final titer (Supplementary Fig. 5b). On the other hand, the highest initial PHB production

rate (Supplementary Fig. 6a) and final yield (Supplementary Fig. 6b) were obtained with the addition of 2 mM TPP. Consequently, all subsequent PHB production experiments using 100 mM substrate were conducted under conditions featuring 2 mM NADP⁺, 0.5 mM CoA, and 2 mM TPP.

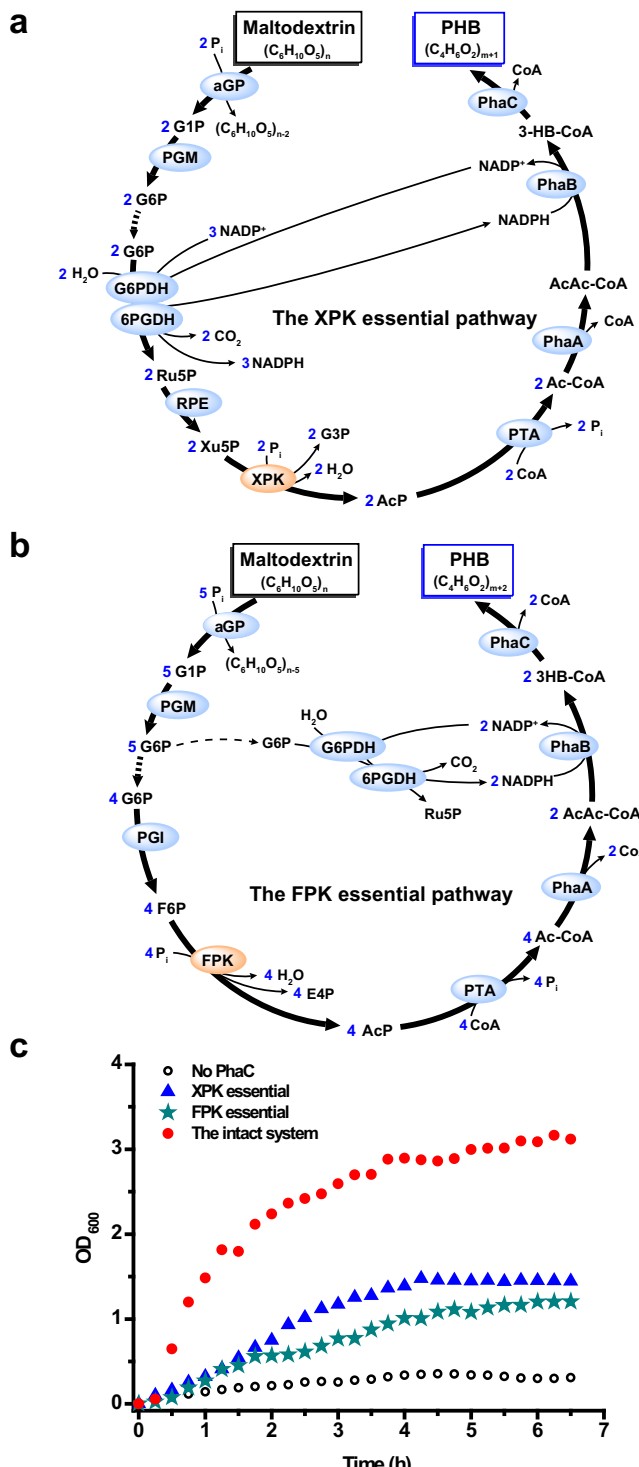

**Fig. 3 | Proof-of-concept production of PHB from debranched maltodextrin by the designed ivSEB. a** Schematic of the XPK essential pathway (i.e. without TK, TIM, ALD, FBP, TAL, RPI, and PGI), in which the consumption of 2 glucose equivalents of maltodextrin results in the generation of 1 monomer equivalent of PHB. **b** Schematic of the FPK essential pathway (i.e. without TK, TIM, ALD, FBP, TAL, RPI, and RPE), in which the consumption of 5 glucose equivalents of maltodextrin results in the generation of 2 monomer equivalents of PHB. **c** Results showing the production of PHB from around 55.6 mM glucose equivalent (10 g/L) of maltodextrin. The reactions commenced upon the addition of maltodextrin and were continuously monitored at 15 min intervals via absorbance at 600 nm ($OD_{600}$) throughout the experiment. Source data are provided as a Source Data file.

## In silico optimization of enzyme concentrations for PHB production

Following the outcomes obtained with five-fold enzyme concentrations, we conducted a preliminary investigation aimed at reducing the enzyme loadings while ensuring comparable system performance. Given the complexity of our ivSEB, composed of 17 enzymes, optimizing enzyme concentrations experimentally by systematic titration[32] would be laborious. Therefore, we made modifications to the COPASI model, primarily by altering the kinetic functions and parameters, rendering it semi-quantitative to facilitate the in silico enzyme optimization (see Methods). The unfitted model (Model 0, with all kinetic parameters summarized in Supplementary Table 5) predicted significantly lower PHB titers, slower reaction rates, and less maltodextrin consumption compared with experimental observations (Supplementary Fig. 7a, b). By reducing the $K_m$ values of PKL and TK for substrates within the model, the final PHB titer of the simulation reaction increased noticeably, but the rate of the simulation reaction remained slow. Then it was observed that the decrease of $K_m$ values of PTA and PhaB for substrates promoted the simulation reaction rate. Finally, the parameters of PhaC were adjusted to make the PHB titers consistent with the experimental values (Supplementary Fig. 7c, d), resulting in the fitted Model 1 (kinetic parameters are summarized in Supplementary Table 5). Using Model 1, simulative enzyme optimization was performed by running the Parameter Scan task in COPASI. The $V_{max}$ values of enzymes in the model, corresponding to enzyme loading concentrations for the experiments, were scanned one at a time. The first round of scanning suggested that adjusting the concentration of PhaC may enhance the PHB titer (Supplementary Fig. 8a). When $V_{max}$ of PhaC increased to 8 mM/min (8 U/mL for the enzyme loading amount), 79.8 mM PHB was expected to be produced from 60.0 mM maltodextrin, achieving a theoretical molar yield (Supplementary Fig. 8b). Then, at this optimized $V_{max}$ value of PhaC, Parameter Scan was carried out again for the rest of enzymes to check whether enzyme concentrations could be reduced without significant negative impact on the initial reaction rate. The scanning results, as displayed in Supplementary Fig. 8c–r, indicated PKL is a rate-limiting enzyme requiring higher concentration for effective reaction rates. PKL was also considered as a potential bottleneck in the glucose-based ivSEB[28]. In contrast, G6PDH might exhibit a kinetic trap scenario[33], displaying a negative relationship between its concentration and the initial PHB production rate, and recommending a reduced concentration.

Guided by the in silico optimization results derived using Model 1 (Supplementary Fig. 8), a subsequent trial of the maltodextrin-to-PHB experiment was carried out employing the predicted optimal enzyme concentrations (summarized in Supplementary Table 6). The total enzyme concentration for this trial was 9.5 mg/mL, decreased by approximately 42.3% in terms of mg/mL compared to the trial using five-fold enzyme concentrations (Supplementary Table 6). Under this optimized condition, the actual biosystem reached equilibrium at around 8 h when 55.2 ± 0.8 mM maltodextrin was consumed and 70.4 ± 1.2 mM (6.0 ± 0.1 g/L) PHB was produced (Supplementary Fig. 9). The resulting molar yield of 127.5% surpassed the pre-optimization yield of 117.2%, and was close to the theoretical yield of 133.3%. Nonetheless, this trial exhibited a slower reaction rate and a lower PHB final titer compared with the simulation results (Supplementary Fig. 9).

Recognizing the inherent complexity of our ivSEB and the challenges in developing an accurate model reflecting real-world performance, a second round of model fitting was undertaken to align simulation data with the aforementioned three sets of experimental data. During this process, adjustments were made to the $k_{eq}$ value of αGP and the $C_{bind}$ value of PhaC, leading to the refined Model 2 (kinetic parameters detailed in Supplementary Table 5). Notably, Model 2

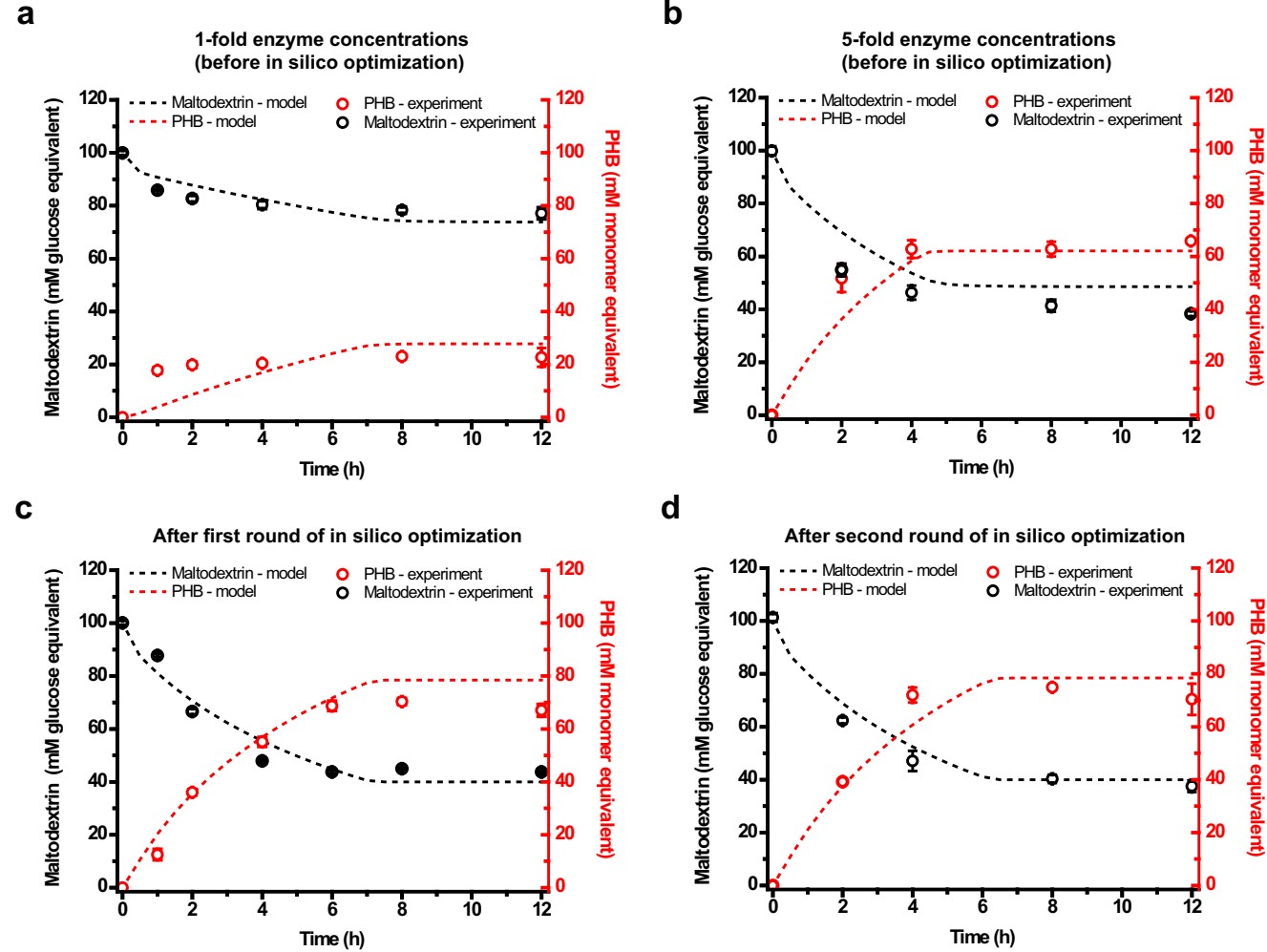

**Fig. 4 | One-pot, one-step production of PHB from 100 mM debranched maltodextrin by the designed ivSEB.** Experimental data were represented as dots, while simulation results were shown as dotted lines, generated using Model 2 which had undergone two rounds of model fitting. The first round of model fitting utilized experimental data from Figs. 4a, b. The second round of model fitting used experimental data from Fig. 4a–c. **a** PhaA was loaded at 1 mg/mL (0.78 mU/mL) while other enzymes were loaded at 1 U/mL. **b** PhaA was loaded at 5 mg/mL (3.9 mU/ mL) while other enzymes were loaded at 5 U/mL. **c** Enzymes were loaded at the suggested concentrations after the first round of in silico optimization, as summarized in Supplementary Table 6. **d** Enzymes were loaded at the suggested concentrations after the second round of in silico optimization, as summarized in Supplementary Table 6. Reactions were performed in triplicate (*n* = 3 biologically independent samples) and data are presented as mean values ± standard deviation (SD). Source data are provided as a Source Data file.

demonstrated enhanced efficacy in simulating the data compared with Model 1 (Fig. 4a–c), particularly for data obtained after the first round of in silico optimization (Fig. 4c and Supplementary Fig. 9). Subsequently, a second round of in silico optimization using Model 2 was conducted (Supplementary Fig. 10 and Supplementary Table 6). Compared with the optimized enzyme concentrations proposed by Model 1, the optimized set predicted by Model 2 mainly differed by an increase in the concentration of αGP and a decrease in the concentrations of PhaB and 6PGDH in terms of mg/mL, while the total enzyme concentration showed minimal divergence (Supplementary Table 6). Guided by the outcomes of this second-round of in silico optimization, the experimental ivSEB achieved an accelerated initial reaction rate, closely aligning with the stimulation outcomes (Fig. 4d). The reaction exhibited rapid kinetics in the first 4 h, followed by a deceleration, reaching equilibrium at approximately 8 h, resulting in the production of 74.9 ± 0.5 mM (6.4 ± 0.04 g/L) PHB from the consumption of 59.7 ± 1.6 mM maltodextrin at a production rate of 9.4 mM/h (approximately 0.8 g/L/h) (Fig. 4d). All of these values surpassed those obtained in the trial using the first-round optimized enzyme concentrations. The resulting molar yield of 125.5% was marginally lower than the first optimized trial (127.5%).

Using the optimized enzyme and cofactor concentrations, we further evaluated the performance of our ivSEB in converting crude starch to PHB. Edible crude starch from four different sources – corn, yam, cassava, and wheat – were pre-treated by IA and added to the reaction system at a final concentration of approximately 80 – 100 mM glucose equivalent. After a 10-h reaction period, production of PHB from these starches ranged from around 26.2 to 35.1 mM (2.3 – 3.0 g/L), correlating to molar yields of 95.7 – 105.9% (Supplementary Fig. 11a, b). These results affirmed the feasibility of our ivSEB for the conversion of crude starch to PHB. However, considering the poor solubility of starch – a potential factor contributing to lower PHB titers compared to soluble maltodextrin – and recognizing that soluble maltodextrin facilitates the separation of insoluble PHB, we intended to persist in employing maltodextrin as the substrate for subsequent investigations, with a specific focus on achieving complete substrate utilization.

**The complete conversion of maltodextrin to PHB**
During PHB production, maltodextrin underwent gradual shortening to maltose, posing difficulties in its further utilization by αGP[14]. This incomplete utilization of maltodextrin within our designed 17-enzyme

ivSEB prompted the incorporation of a second reaction step involving two auxiliary enzymes, 4-α-glucanotransferase (4GT) and polyphosphate glucokinase (PPGK). 4GT catalyzes the transglycosylation among maltooligosaccharides, converting maltose into longer maltodextrin that αGP can utilize[34]. This transglycosylation process also generates glucose, which can be converted to G6P in the presence of polyphosphate (polyP) by PPGK[35]. Importantly, the addition of these two enzymes, especially 4GT, must be sequenced as a second stage, initiated after the 17-enzyme ivSEB reaction attains equilibrium. Premature supplementation of 4GT may lead to reduced maltodextrin utilization efficiency due to its side reaction involving cyclization of long-chain starch/maltodextrin[32,36].

In pursuit of achieving the full conversion of 100 mM maltodextrin into PHB, a one-pot two-step reaction was conducted (Fig. 5a). The first step, spanning from 0 to 8 h, replicated a previous experiment depicted in Fig. 4d, with a modification of adding PhaC at 20 U/mL to ensure the complete conversion of 100 mM substrate to PHB. This adjustment was guided by in silico optimization using Model 2 (Supplementary Fig. 12). Subsequently, the second step commenced at 8 h when 4GT, PPGK, and polyP were added. Experimental optimization of the concentrations of 4GT and PPGK revealed that when 4GT was loaded at 0.2 U/mL and PPGK at 2.0 U/mL (approximately 0.67 mg/mL and 0.02 mg/mL, respectively), the ivSEB showed a better performance (Supplementary Fig. 13). At these optimized enzyme concentrations, the reaction exhibited rapid progress during 8 – 12 h, followed by a slowing down. The near-complete maltodextrin consumption and the highest PHB titer were both achieved at 24 h, yielding $118.8 \pm 1.2$ mM ($10.2 \pm 0.1$ g/L) PHB from the consumption of $98.9 \pm 1.0$ mM maltodextrin, reflecting a PHB molar yield of 120.1%. The entire one-pot, two-step production process is summarized in Fig. 5a. The produced PHB exhibited a weight-average molecular weight ($M_w$) of $2.97 \times 10^5$ and a polydispersity index (PDI) of 1.65 (Supplementary Table 7). This PDI value was comparable to those (1.35 and 1.48) obtained by Satoh et al.[23] using an ivSEB to produce PHB from acetate, and was lower than the PDIs of the other two PHB samples analyzed in parallel, both of which were of microbial origins (Supplementary Table 7).

Furthermore, to investigate the scalability of our ivSEB, we doubled the maltodextrin concentration to 200 mM. Initially maintaining the concentrations of enzymes, cofactors, and phosphate ions as in Fig. 5a, a single addition of 200 mM maltodextrin at 0 h resulted in a sluggish reaction rate and a low PHB product titer during the 24 h reaction period (Supplementary Fig. 14). Suspecting substrate inhibition as the cause for the low reaction rate and PHB titer, we then implemented a fed-batch substrate addition strategy, adding the same amount of maltodextrin at 0 h and 4 h to achieve a final concentration of 200 mM. In contrast to Fig. 5a, where the reaction slowed down after 4 h without additional substrate, this time, the reaction rate during 4 – 8 h was similar to that of 0 – 4 h, suggesting the potential resolution of the substrate inhibition issue. At 12 h, 4GT and PPGK were added at the previously optimized concentrations. The reaction continued until 24 h, reaching an equilibrium with the production of $202.8 \pm 3.1$ mM ($17.4 \pm 0.3$ g/L) PHB from the consumption of $185.8 \pm 0.5$ mM maltodextrin, corresponding to a PHB molar yield of 109.1% and an overall reaction rate of 8.5 mM/h (approximately 0.7 g/L/h) (Fig. 5b). To simplify the handling process, doubling the concentrations of all enzymes (including auxiliary enzymes), cofactors, and phosphate ions, along with a single addition of 200 mM maltodextrin at the beginning, resulted in an enhanced reaction rate during the first 4 h (Fig. 5c). Followed by the addition of 4GT and PPGK at 8 h, the reaction proceeded at a relatively constant rate till 24 h, when $208.3 \pm 10.0$ mM ($17.9 \pm 0.9$ g/L) PHB was produced from the consumption of $186.1 \pm 2.6$ mM maltodextrin. This strategy yielded similar results to those of the fed-batch experiment of Fig. 5b with a slightly enhanced molar yield of 111.9% and a slightly increased overall reaction rate of 8.7 mM/h (approximately 0.7 g/L/h) (Fig. 5c). Compared with previously reported PHB-producing ivSEBs, our system not only provided a straightforward operational process for producing PHB from starch-derived maltodextrin, but also achieved the near-theoretical product yield, the highest product titer, and the fastest reaction rate (Table 1), indicating that this study laid a solid foundation for scaling up the biosynthesis of acetyl-CoA-derived products from starch in an industrial setting.

## Discussion

The iterative development process of ivBT follows a design-build-test-learn framework involving pathway design, enzyme selection, enzyme production, and process engineering efforts[3]. As the cornerstone of ivBT development, pathway design requires considerations including substrate costs, coenzyme and ATP balance, reaction equilibrium, and thermodynamics[3,11,37]. In this study, we focused on the development of an efficient ATP-free and NADP(H)-balanced ivSEB that rapidly converts starch/maltodextrin into PHB with high yield and titer. Compared with the glucose-based ivSEB reported by Opgenorth et al.[28], our ivSEB presented several key enhancements. Firstly, our ivSEB exclusively used starch or maltodextrin as substrate to generate phosphate sugars with $P_i$, bypassing the need for costly and unstable ATP. Secondly, our ivSEB replaced PfkB-catalyzed ATP regeneration ($\Delta_r G^{\circ} = 16.9 \pm 1.3$ kJ/mol) in the glucose-based system with the energy-favorable FBP-catalyzed $P_i$ regeneration ($\Delta_r G^{\circ} = -12.4 \pm 1.4$ kJ/mol), which may partly account for the much higher PHB production rate of our system (Table 1). Thirdly, our pathway eliminated the need for NAD(P)H purge valves, validated by the model-based stoichiometric analysis and quantitative experiments. Fourthly, our process achieved a notably enhanced PHB molar yield of around 120% from 100 mM glucose equivalent of substrate, compared to the approximately 85% yield reported in the glucose-based ivSEB. Lastly, our process simplified operations by requiring a single PhaC loading at the outset, thereby eliminating the need for multiple steps involving product removal and PhaC re-addition. Importantly, in contrast to the challenges associated with separating PHB from microbial cells through fermentation[38], our ivSEB facilitates the easy separation of insoluble PHB produced. While there may be room for further improvements, this study reveals the industrial potential of our designed ivSEB for the high-yield production of PHB from starch-derived substrates. Additionally, our designed pathway could be adapted to synthesize other acetyl-CoA-derived products, such as n-butanol, lipids, and isoprenoids by altering downstream enzymes.

Stoichiometric analysis is a crucial concept in metabolic engineering, facilitating the evaluation of metabolite fluxes and description of the mass balance of metabolic network[39]. Despite being beneficial to in vitro biosystems as well, its importance is sometimes overlooked due to perceived simplicity of ivBT pathways compared to in vivo metabolic networks. Nonetheless, there has been a recent trend to develop complicated ivSEBs containing 10 to 20 or more enzymes in one pot[10,40–42]. Such pathways may include branch points, making stoichiometric analysis even more intricate. Therefore, it is more reliable and time-saving to perform stoichiometric analysis of the designed pathway with a computational model, which can be readily constructed using an open-source software such as COPASI without precise kinetic data. Our analysis of the glucose-based ivSEB for PHB production[28] revealed similar overall stoichiometric equation to that of our system, written as $3\ C_6H_{12}O_6 = 4\ (C_4H_6O_2) + 2\ CO_2 + 6\ H_2O$, rather than $C_6H_{12}O_6 + 3\ NADP^+ = C_4H_6O_2 + 2\ CO_2 + 3\ NADPH + 3\ H^+$ (the latter is written based on the description in the reference[28]; $C_6H_{12}O_6$ represents glucose), indicating no excessive NADPH production or need for an NAD(P)H purge valve. Estimated theoretical PHB molar yield of this glucose-based ivSEB was therefore revised to 133.3% instead of 100.0%, rectifying an earlier underestimation in the previous study[28] possibly due to the overlook of a pathway branch point that G6P can be used

## a

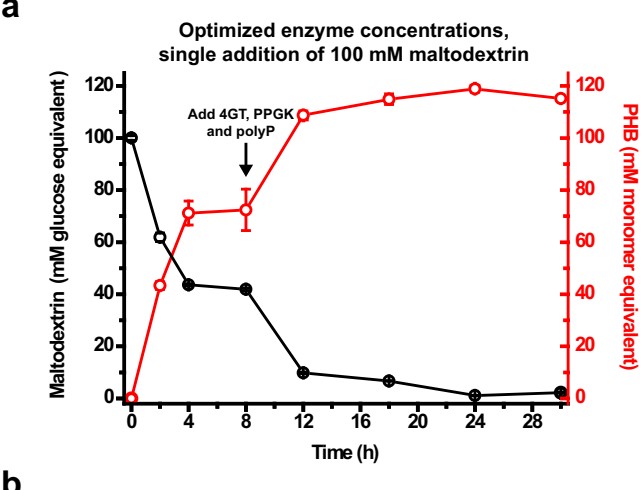

## b

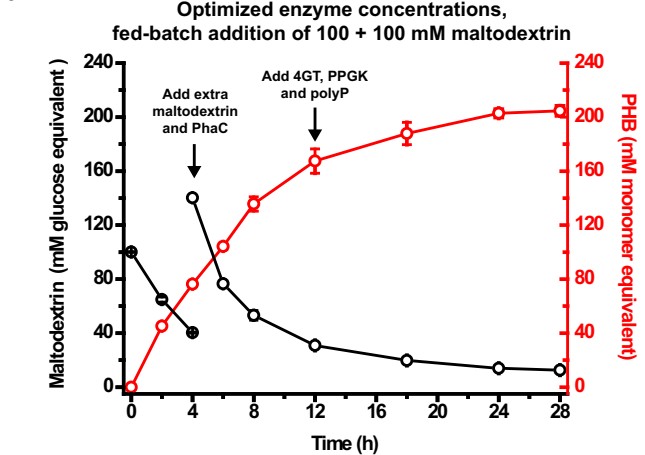

## c

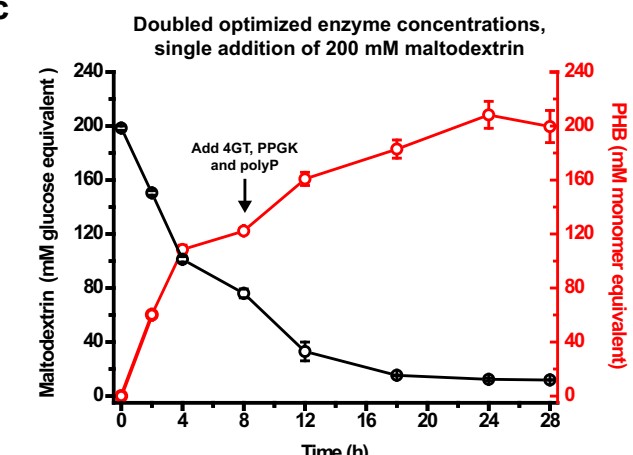

**Fig. 5 | One-pot, two-step production of PHB aiming at achieving complete maltodextrin utilization. a** Production of PHB from 100 mM glucose equivalent of maltodextrin. Initial reaction conditions were the same as those of Fig. 4d (refer to Methods for details), except that PhaC was added at 20 U/mL as suggested by Model 2 (see Supplementary Fig. 12). At 8 h, 4GT, PPGK, and polyphosphate (polyP) in the form of sodium hexametaphosphate were added to the system, reaching final concentrations of 0.2 U/mL (approximately 0.67 mg/mL), 2 U/mL (approximately 0.02 mg/mL), and 20 mM, respectively. **b** Production of PHB from 200 mM glucose equivalent of maltodextrin using a fed-batch substrate addition strategy. Initial reaction conditions aligned with those of Fig. 5a. At 4 h, another 100 mM maltodextrin and 20 U/mL PhaC was added. At 12 h, 4GT, PPGK, and polyP were added to the system at the same concentrations as in Fig. 5a. **c** Production of PHB from 200 mM glucose equivalent of maltodextrin with a single substrate addition. Enzymes, NADP$^+$, CoA, TPP, and phosphate ion concentrations were doubled compared to those in Fig. 5a, b. At 8 h, 4GT, PPGK, and polyP were added to the system at doubled concentrations as in Fig. 5a, b. Reactions were performed in triplicate ($n = 3$ biologically independent samples) and data are presented as mean values ± SD. Source data are provided as a Source Data file.

holds promise for optimizing our system based on a substantial amount of precise data. Secondly, reducing enzyme loading concentrations is crucial to decrease the enzyme cost of ivBT. Beyond optimizing enzyme concentrations, enhancing the activity and stability of enzymes involved in our ivSEB by protein engineering (aided by machine learning algorithms[45]) and gene mining[46] is another important direction. This is particularly pertinent for PhaC, given its binding to the produced PHB[28,47], and for αGP, PhaA, and PhaB, which exhibited relatively low specific activity values and therefore required larger loading amounts in our study. Thirdly, the maltodextrin derived from the established industrial process of starch liquefaction[48] may serve as a direct substrate for in vitro debranching and PHB production in the future. Fourthly, molecular weight analysis revealed that our PHB product exhibited a smaller $M_w$ and lower PDI than two microbial-originated PHB samples, implying its suitability for applications such as tissue engineering scaffolds or drug delivery carriers, aligning with previous investigations on PHB with similar molecular weights[49,50]. Exploring in vitro PHB synthesis for controlled molecular weights and low PDIs will be of great importance.

In summary, we herein propose an ivSEB design using starch or starch derivatives for PHB biosynthesis via acetyl-CoA. Our system offers multiple advantages, including ATP independence, self-driven and self-regulating capabilities (because starch/maltodextrin is the sole substrate, and there is no need for coenzyme regulation modules), high product yields and titers, rapid reaction rates, simple operation, and promising scalability. Our study highlights the significance of pathway design and analysis for achieving high-yield ivSEBs, and demonstrates the significant potential of ivBT platform for the efficient biosynthesis of PHB and other acetyl-CoA-derived products.

## Methods

### Reagents

Unless otherwise specified, all chemicals were of reagent grade or higher and were purchased from Sigma-Aldrich (St. Louis, MO, USA), Sinopharm (Shanghai, China), Aladdin (Shanghai, China), or Solarbio (Beijing, China). PrimeSTAR Max DNA Polymerase from Takara (Tokyo, Japan) was used for the PCR reactions. Primers for PCR were synthesized by Azenta (Suzhou, China). Maltodextrin with a dextrose equivalent (DE) of 4.0 −7.0 (purchased from Sigma-Aldrich, product catalog number 419672) was used in this study. PHB sample 1 for molecular weight analysis was purchased from Sigma-Aldrich (product catalog number 363502), and was also used as a standard for the quantification study. PHB sample 2 for molecular weight analysis was produced by microbial fermentation, and was kindly provided by Xuejun Chen from Tianjin Institute of Industrial Biotechnology, Chinese Academy of Sciences.

not only by G6PDH but also by PGI. Our study highlights the critical role of stoichiometric analysis in complex ivSEBs, offering an accessible method for future studies.

To further improve the efficiency of our ivSEB at an industrial scale, future strategies should be implemented. Firstly, despite the effective application of our semi-quantitative model in facilitating two-round in silico optimization of enzyme concentrations in this study, it is acknowledged that constructing precise kinetic models for complex ivSEBs presents challenges. Diverse factors, including mass transport, protein-protein interactions, substrate/product inhibition, and enzyme/cofactor degradation, can impact enzymatic reaction rates[13,41,43]. Recent research has successfully employed machine learning to optimize enzyme concentrations in ivSEBs[44], and this method

**Table 1 | A comparison of different ivSEBs for PHB production**

| Source of carbon | Need of ATP | Source of ATP | Source of NADPH | NADPH regeneration method | Theoretical system performance | | Reaction mode | Substrate concentration | Actual system performance | | | | Ref. |
|---|---|---|---|---|---|---|---|---|---|---|---|---|---|
| | | | | | Carbon conversion efficiency | Molar yield of PHB | | | Final product titer | Molar yield of PHB | Total reaction time | PHB production rate | |
| Acetate | Yes | Direct addition | Glucose | GDH | 100.0% | 50.0% | One pot, one step | 30.0 mM | 9.3 mM[a] (0.8 g/L) | 31.0% | 24 h | 0.4 mM/h (<0.1 g/L/h) | 23 |
| | | | | | | | One pot, one step | 60.0 mM | 12.8 mM[a] (1.1 g/L) | 21.3% | 24 h | 0.5 mM/h (<0.1 g/L/h) | |
| Acetate | Yes | ATP regeneration by thylakoid membranes | Light | Thylakoid membrane | 100.0% | 50.0% | One pot, one step | 10.0 mM | 4.3 mM (0.4 g/L)[b] | 43.0% | Around 34 h | Around 0.1 mM/h (<0.1 g/L/h) | 24 |
| | | | | | | | One pot, five steps[c] | 50.0 mM | 20.0 mM (1.7 g/L)[b] | 40.0% | 36 h | 0.6 mM/h (<0.1 g/L/h) | |
| Methanol | No | N/A | Methanol | FDH | 80.0% | 20.0% | One pot, one step | 20.0 mM | 3.9 mM (0.3 g/L)[b] | 19.5% | 6.5 h | 0.6 mM/h (<0.1 g/L/h) | 26 |
| | | | | | | | Two pots, three steps[d] | 20.0 mM | 69.8 mM[a] (6.0 g/L) | 20.5% | Not clear[e] | Cannot be calculated | |
| Pyruvate | No | N/A | Pyruvate | PDH (with a purge valve) | 66.7% | 50.0% | One pot, one step | 50.0 mM | 29.1 mM[a] (2.5 g/L) | 58.2% | 15 h | 1.9 mM/h (0.2 g/L/h) | 27 |
| Glucose | Yes | ATP regeneration from glucose | Glucose | G6PDH, 6PGDH (with purge valves) | 88.9%[f] | 133.3%[f] | One pot, three steps[g] | 60.7 mM | 56.8 mM (4.9 g/L)[b] | 93.6% | 30 h | 1.9 mM/h (0.2 g/L/h) | 28 |
| | | | | | | | One pot, five steps[g] | 109.2 mM | 93.8 mM (8.1 g/L)[b] | 85.9% | 55 h | 1.7 mM/h (0.1 g/L/h) | |
| Starch-derived maltodextrin | No | N/A | Starch-derived maltodextrin | G6PDH, 6PGDH (no purge valve) | 88.9% | 133.3% | One pot, one step | 101.4 mM | 74.9 mM (6.4 g/L)[b] | 125.5% | 8 h | 9.4 mM/h (0.8 g/L/h) | This study |
| | | | | | | | One pot, two steps | 100.0 mM | 118.8 mM (10.2 g/L)[b] | 120.1% | 24 h | 5.0 mM/h (0.4 g/L/h) | |
| | | | | | | | One pot, two steps | 198.4 mM | 208.3 mM (17.9 g/L)[b] | 111.9% | 24 h | 8.7 mM/h (0.7 g/L/h) | |

[a] Results provided in mM were calculated from the data in g/L based on the molecular weight (86) of the monomer of PHB.
[b] Results provided in g/L were calculated from the data in mM based on the molecular weight (86) of the monomer of PHB.
[c] In the first step, 10 mM substrate was added. In each of the remaining steps, PHB was removed from the reaction mixture, followed by the addition of PhaC.
[d] The three steps are (1) the conversion of 20 mM methanol to glycolaldehyde by a 1-L system; (2) the concentration of the reaction solution to 58.8 mL, which contained 135.6 mM glycolaldehyde, and (3) the conversion of concentrated glycolaldehyde to PHB.
[e] The timings of the first two steps were not provided in the reference. The reaction time of the third step was 5 h.
[f] Theoretical carbon conversion efficiency provided by the reference article was 66.6%, corresponding to a theoretical PHB molar yield of 100.0%. However, we suggest these numbers be 88.9% and 133.3%, respectively. Explanation refers to the Discussion section.
[g] In the first step, the reaction was initiated by a single addition of substrate. In each of the remaining steps, PHB was removed from the reaction mixture by centrifugation, followed by the addition of PhaC.

## Construction of plasmids

Plasmids pET28a-CnPhaA for the expression of PhaA from *Cupriavidus necator* H16 (UniProt ID. P14611) and pET28a-CnPhaB for the expression of PhaB from *C. necator* H16 (UniProt ID. P14697) were constructed by Simple Cloning[51]. Each insertion fragment containing the target gene was amplified from the bacterial genome by regular PCR with a pair of primers IF and IR, and the pET28a vector fragment was amplified from the pET28a vector (Novagen, Madison, WI, USA) with a pair of primers VF and VR (Supplementary Table 8). The insertion and vector fragments were then concomitantly used as primers and templates for prolonged overlap extension PCR (POE-PCR) to generate DNA multimer. Competent *E. coli* TOP10 (CWBio, Beijing, China) was transformed with the DNA multimer to yield the desired plasmid. Plasmid pET28a-CsPhaC for the expression of PhaC from *Cupriavidus sp*. S-6 (UniProt ID. G8BLJ2) was constructed by Azenta (Suzhou, China). The gene encoding PhaC was codon-optimized and inserted into pET28a vector between NdeI and XhoI restriction sites. Plasmids for the expression of αGP[40], PGM[52], PGI[53], PKL[40], PTA[54], G6PDH (wild-type)[55], 6PGDH (wild-type)[56], the carbon rearrangement enzymes (TAL, TK, RPI, RPE, TIM, ALD, FBP)[40], and the auxiliary enzymes for the complete utilization of maltodextrin (4GT[36], engineered PPGK mutant 4-1[35]) were prepared as previously described. The plasmid sequences were validated using DNA sequencing by Azenta (Suzhou, China).

## Protein expression and purification

*E. coli* BL21(DE3) (Invitrogen Co., Carlsbad, CA, USA) was used for recombinant protein expression. Each enzyme was expressed individually. Cells transformed with plasmid were plated on Luria-Bertani (LB) agar with 100 µg/mL ampicillin or 50 µg/mL kanamycin, and incubated overnight at 37 °C. Colonies were inoculated into LB medium with either 100 µg/mL ampicillin or 50 µg/mL kanamycin in a shake flask, cultivated at 37 °C and 250 rpm till the absorbance of the cell culture at 600 nm ($OD_{600}$) reached 0.8 – 1.0. Recombinant protein expression was then induced by adding isopropyl-β-D-thiogalactopyranoside (IPTG) to a final concentration of 0.1 mM. The bacterial culture was further incubated at 18 °C for 20 h. The cells were harvested by centrifugation at 4 °C, washed once with 100 mM HEPES containing 250 mM NaCl (pH 7.4), resuspended in the same buffer to a final $OD_{600}$ of 60–80, and lysed by sonication. After centrifugation, the target enzymes in the supernatants were purified. PGM and PGI were purified by affinity adsorption of carbohydrate-binding module (CBM) on regenerated amorphous cellulose followed by self-cleavage of intein as previously reported[52,53]. TIM was purified by incubating the crude cell lysate in a water bath at 70 °C for 30 min followed by centrifugation. The other enzymes were purified by nickel-affinity chromatography using Ni Sepharose 6 Fast Flow medium (GE Healthcare, UK). The purities of the recombinant enzymes were examined by SDS–PAGE. Protein concentrations were determined by the Bradford method using bovine serum albumin as standard.

## Enzymatic activity assays

Enzyme activities were determined at 37 °C in 200 mM Tris-HCl buffer (pH 7.4). One unit (U) of enzyme activity was defined as the amount of enzyme that consumed 1 µmole of substrate or generated 1 µmole of product per min. PKL activity was measured in buffer containing 5 mM $MgCl_2$, 50 mM sodium phosphate (pH 7.4), 0.5 mM thiamine pyrophosphate (TPP), 5 mM F6P or Xu5P. The amount of AcP produced was determined using the colorimetric hydroxamate assay[57]. PTA activity was measured in buffer containing 5 mM $MgCl_2$, 5 mM sodium phosphate (pH 7.4), 0.01 mM TPP, 0.1 mM CoA, 1 mM F6P, and 1 U/mL purified PKL (in terms of FPK activity). The reaction was initiated by adding PTA, and the formation of acetyl-CoA was monitored at 233 nm ($\varepsilon_{233}$ = 4.44 mM$^{-1}$cm$^{-1}$, ref. [58]) by a Cary 100 UV-Vis spectrophotometer (Agilent Technologies, USA). PhaA activity assay was carried out in buffer containing 5 mM $MgCl_2$ and 0.15 mM acetyl-CoA. The reaction was initiated by adding PhaA, and the formation of acetoacetyl-CoA was monitored at 303 nm ($\varepsilon_{303}$ = 12.9 mM$^{-1}$cm$^{-1}$, ref. [23]). PhaB activity assay was carried out in buffer containing 5 mM $MgCl_2$, 0.2 mM NADPH, 0.1 mM acetyl-CoA, and 5 mg/mL purified CnPhaA. The reaction was initiated by adding PhaB, and the decrease of NADPH was monitored at 340 nm ($\varepsilon_{340}$ = 6.22 mM$^{-1}$cm$^{-1}$). The activity of G6PDH was measured in buffer containing 5 mM $MgCl_2$, 0.5 mM $MnCl_2$, 1 mM G6P, and 0.1 mM NADP$^+$. The reaction was initiated by adding G6PDH, and the production of NADPH was monitored at 340 nm. The activity of 6PGDH was measured in buffer containing 5 mM $MgCl_2$, 0.5 mM $MnCl_2$, 1 mM 6-phosphogluconate, and 0.02 mM NADP$^+$. The reaction was initiated by adding 6PGDH, and the production of NADPH was monitored at 340 nm. Specific activities of 4GT and PPGK were determined as previously described[32].

## Computational modeling

Construction of the computational model and the related analysis were conducted using CopasiUI (version 4; https://copasi.org/)[29]. A fixed compartment was created, and 19 reaction equations were entered as displayed in Supplementary Table 4. For stoichiometric analysis, kinetic functions except that for PhaC (Supplementary Table 3, functions 1- 5) were obtained from previous studies[7,59], and the Henri-Michaelis-Menten function (Supplementary Table 3, function 6) was used for PhaC. Values of all kinetic parameters ($k_{eq}$, $K_m$, and $V_{max}$) were randomly set as 1. The units of $K_m$ and $V_{max}$ were mM and µmol/(mL·min) (i.e. mM/min), respectively. Species types were set as "fixed" for maltodextrin and PHB, and "reactions" for the rest of contents in the system. Then, a Stoichiometric Analysis task using the Elementary Modes was carried out to calculate the stoichiometric profiles of the designed ivSEB.

For the construction of a semi-quantitative kinetic model, the above-mentioned model was further adjusted. Species type was set as "reactions" for all components in the system. For initial settings, $k_{eq}$ values listed in Supplementary Table 2 were calculated using eQuilibrator 3.0 (https://equilibrator.weizmann.ac.il/)[60], with pH set as 7.4, pMg = 2, and ionic strength = 0.2 M. $K_m$ values were mostly obtained from the BRENDA database (https://www.brenda-enzymes.org/), and $K_m$ values that could not be obtained from any references remained at 0.1 mM initially (Supplementary Table 5). Based on the experimental result that around 60% of the substrate could be consumed at equilibrium (Supplementary Fig. 3b), an event was set so that the simulation reaction ceased when the transient concentration of maltodextrin in the system was no more than 40% of the initial maltodextrin concentration. Considering that PhaC binds to and co-precipitates with PHB during the reaction[28,47], a specific function for PhaC (Supplementary Table 3, function 7) was developed based on the Henri-Michaelis-Menten equation, with the addition of a binding coefficient ($C_{bind}$) as well as an exponent of PHB concentration (n) whose initial values were randomly set as 1000 and 1, respectively. The kinetic model using these initial parameters values is named as Model 0.

For the first round of model fitting, two sets of previously obtained experimental data (as displayed in Supplementary Fig. 3b, d) were used. Same as in the experiment, initial concentrations of maltodextrin, $P_i$, CoA, and NADP$^+$ in the model were set as 100, 10, 0.5, and 2 mM, respectively. Consistent with enzyme loading concentrations in these two sets of experiments, $V_{max}$ of all enzymes except FPK in the model were set as the same numerical value, being 1 or 5 mM/min. During model fitting, $V_{max}$ values of XPK and FPK in our model constantly followed a fixed ratio of 1:0.12 based on the specific activity values of PKL for Xu5P and F6P (shown in Supplementary Table 1). $K_m$, $C_{bind}$, and n values were adjusted to fit the model to experimental data, resulting in Model 1.

The second round of model fitting followed the same procedures as the first round of model fitting, except that three sets of previously obtained experimental data (as displayed in Supplementary Fig. 3b, d,

and Supplementary Fig. 9) were used. $K_m$, $k_{eq}$, $C_{bind}$, and n values were adjusted to fit the model to experimental data, resulting in Model 2.

## In silico optimization of enzyme loading concentrations

Two rounds of in silico optimization of enzyme loading concentrations were performed using Model 1 and Model 2, respectively.

For the first round of in silico optimization using Model 1, the $V_{max}$ values of all enzymatic reactions except that of FPK (as explained above in the Computational modeling section) were set as 5 mM/min. Each enzyme was scanned individually for its optimal $V_{max}$ value, while the $V_{max}$ values of other enzymes remained unchanged. For the bifunctional PKL, XPK and FPK were scanned simultaneously at a fixed $V_{max}$ ratio of 1:0.12 as explained above. Similarly, for the bifunctional TK, TK-1 and TK-2 reactions (specified in Supplementary Table 2) were scanned simultaneously at a fixed $V_{max}$ ratio of 1:1. At first, $V_{max}$ of each enzyme was scanned individually within the range of 1 to 10 mM/min over a timescale of 240 min to find enzymes whose concentrations significantly affected the final PHB yield. This timescale was determined based on the experimental results shown in Supplementary Fig. 3d. Then, $V_{max}$ of PhaC was adjusted to 8 mM/min, and the rest of enzymes were scanned individually again within the $V_{max}$ range of 1 to 5 mM/min over a timescale of 60 min to determine their optimal $V_{max}$ values based on initial PHB production rate. Initial PHB production rate was defined as the amount of PHB produced within the first h of simulation reaction. Optimal $V_{max}$ was defined as the minimal $V_{max}$ value that resulted in no less than 99% of the highest initial PHB production rate achieved within the range of $V_{max}$ for scanning, and was converted to enzyme loading concentration (in mg/mL for PhaA, and in U/mL for the other enzymes) at a 1:1 numerical ratio to guide the actual experiments.

The second round of in silico optimization using Model 2 generally followed the same method as described above. But to begin with, the $V_{max}$ values of all enzymatic reactions were set as their optimal values predicted by Model 1. In addition, to scan for the optimal $V_{max}$ of PhaC, the timescale of scanning was set to be 480 min based on the experimental results shown in Supplementary Fig. 9.

## Production of PHB from maltodextrin

Maltodextrin debranched by isoamylase (IA) was used as the substrate for the production of PHB. Methods for IA treatment and quantification of maltodextrin were described previously[32]. 1 mM maltodextrin refers to 1 mM glucose equivalent of maltodextrin. Proof-of-concept experiment was carried out at 37 °C in a 4-mL reaction mixture containing 200 mM Tris-HCl (pH 7.4), 10 mM MgCl$_2$, 0.5 mM MnCl$_2$, 10 μg/mL ampicillin, 5 μg/mL kanamycin, 10 mM sodium phosphate (pH 7.4), 0.5 mM TPP, 0.5 mM CoA, 2 mM NADP$^+$, 10 g/L (around 55.6 mM glucose equivalent) IA-debranched maltodextrin, and enzymes. PhaA was loaded at a final concentration of 1 mg/mL (0.78 mU/mL), while each of the rest of the enzymes was loaded at a final concentration of 1 U/mL. The absorbance of the reaction mixture at 600 nm (OD$_{600}$) was measured in real-time using a Cary 100 UV-Vis spectrophotometer (Agilent Technologies, USA). Unless specified, one-pot, one-step production of PHB from 100 mM maltodextrin was conducted at 37 °C in a 1-mL reaction mixture containing 200 mM Tris-HCl (pH 7.4), 10 mM MgCl$_2$, 0.5 mM MnCl$_2$, 10 μg/mL ampicillin, 5 μg/mL kanamycin, 10 mM sodium phosphate (pH 7.4), 0.5 mM TPP, 0.5 mM CoA, 2 mM NADP$^+$, 100 mM IA-debranched maltodextrin, and enzymes. At different time points, a 0.1-mL aliquot was collected, and centrifuged at 8,000×g to harvest the pellet for PHB quantification.

For the complete utilization of maltodextrin, the reaction was performed in a one-pot, two-step mode. When 100 mM IA-debranched maltodextrin was used as the substrate, the first step was initiated under the abovementioned conditions at optimized enzyme concentrations predicted by Model 2 (displayed in Supplementary Table 6) and coenzyme concentrations (2 mM TPP, 0.5 mM CoA, 2 mM

NADP$^+$), except that the concentration of PhaC was raised to 20 U/mL. After 8 h, 4GT and PPGK were added to the reaction mixture at either 1-fold loading concentrations (0.1 U/mL 4GT, 1.0 U/mL PPGK) or 2-fold loading concentrations (0.2 U/mL 4GT, 2.0 U/mL PPGK) together with 20 mM sodium hexametaphosphate for further PHB production. For the complete utilization of 200 mM maltodextrin using a fed-batch substrate addition strategy, the initial reaction conditions were the same as those using 100 mM maltodextrin. At 4 h, another 100 mM maltodextrin and another 20 U/mL PhaC was added. At 12 h, 4GT, PPGK, and sodium hexametaphosphate were added to the system at 0.2 U/mL (approximately 0.67 mg/mL), 2 U/mL (approximately 0.02 mg/mL), and 20 mM, respectively. For the complete utilization of 200 mM maltodextrin upon a single addition of substrate at 0 h, the concentrations of substrate, MgCl$_2$, MnCl$_2$, sodium phosphate, TPP, CoA, NADP$^+$, and enzymes were all two times those in the trial using 100 mM substrate. After 8 h, 40 mM sodium hexametaphosphate, 0.4 U/mL (approximately 1.33 mg/mL) 4GT, and 4.0 U/mL (approximately 0.04 mg/mL) PPGK were added to the reaction mixture for further PHB production. At different time points, samples were collected for quantification of maltodextrin and PHB. Residual maltodextrin in the sample supernatant was quantified by the total starch assay kit (Megazyme, Ireland) as instructed.

## Production of PHB from crude starch

Edible crude starch from four different sources were used for the production of PHB. Corn starch was purchased from Shanghai Fengwei Industrial Co., Ltd. (348 Gongyuan Road, Qingpu District, Shanghai). Yam starch was purchased from Chongqing Jiaxian Food Co., Ltd. (No. 488 Baohuan Road, Baoshenghu Street, Yubei District, Chongqing, China). Cassava starch and wheat starch were purchased from Nanjing Ganzhiyuan Sugar Co., Ltd. (733 Sheng'an Avenue, Binjiang Development Zone, Jiangning District, Nanjing, China). Starch was debranched by IA prior to use, employing the same procedure as that utilized for the debranching of maltodextrin, as described previously[32]. One-pot, one-step production of PHB from starch was conducted at 37 °C in a reaction mixture containing 200 mM Tris-HCl (pH 7.4), 10 mM MgCl$_2$, 0.5 mM MnCl$_2$, 10 μg/mL ampicillin, 5 μg/mL kanamycin, 10 mM sodium phosphate (pH 7.4), 2 mM TPP, 0.5 mM CoA, 2 mM NADP$^+$, around 80 − 100 mM glucose equivalent of IA-debranched starch. Enzymes were loaded at the optimized concentrations predicted by Model 2 (displayed in Supplementary Table 6). At different time points, samples were collected for the quantification of starch and PHB. Residual starch in the sample was quantified by the total starch assay kit (Megazyme, Ireland) as instructed.

## Quantification of PHB

PHB was quantified by gas chromatography (GC) as described previously[28] with some modifications. Insoluble contents collected from the ivSEB were first dried at 60 °C in an oven for 4 h. Then, 1 mL chloroform and 1 mL methanolysis reagent (prepared by mixing 85 mL methanol, 15 mL concentrated sulfuric acid, and 0.7 g benzoic acid) were added to each dried pellet. The mixture was heated at 95 °C for 4 h in a screw-cap glass tube, followed by cooling to room temperature. Subsequently, 1 mL deionized water was added for extraction and phase separation. The bottom organic layer was passed through an organic filter membrane and used for the quantification of PHB by a GC7900 gas chromatograph system (Techcomp, China) equipped with a J&W HP-5 GC column (30 m × 0.32 mm inner diameter × 0.25 μm film thickness; Agilent Technologies, USA) and an FID detector. Nitrogen was used as a carrier gas. The following temperature program was used: oven temperature was initially maintained at 80 °C for 1.5 min, then increased by 30 °C/min to 140 °C, followed by an increase of 40 °C/min to 240 °C, and maintained at 240 °C for 2 min. Temperatures for the injector and detector were 130 and 240 °C, respectively. The sample injection volume was 2 μL. The peak representing methyl

3HB in a sample was determined using a PHB standard (Sigma) that underwent methanolysis via the same method, and the peak area was compared with that of methyl benzoate (produced from acidic methanolysis of benzoic acid which was used as an internal standard) in each sample for quantification. Data were the mean ± standard deviation of triplicated samples. 1 mM PHB refers to PHB containing 1 mM 3HB unit. The molar yield of PHB was calculated based on the amount of maltodextrin consumed. PHB titers provided in g/L were calculated from the results in mM based on the molecular weight (86) of the monomer of PHB.

All plotted data in the Figures and Supplementary Figs. was analyzed using Microsoft Excel 2019 or OriginPro (version 8). Reaction pathways were drawn using ChemBioDraw (version 14.0). Multi-panel Figures were arranged using Adobe Illustrator CC 2014.

### Molecular weight analysis

PHB was collected from the reaction system by centrifugation, followed by washing twice with deionized water, and extraction using chloroform. The molecular weights of PHB samples were determined by gel permeation chromatography (GPC) on a PL-GPC50 apparatus (Agilent Technologies, USA) equipped with a PL gel 10 μm Mixed-B column coupled in series with a PL gel 10 μm Mixed-D column (Agilent Technologies, USA). Samples were eluted with chloroform at a flow rate of 1.0 mL/min at 35 °C. Polystyrenes (Agilent Technologies, USA) with different molecular weights were used as standards. Weight-average molecular weight ($M_w$), number-average molecular weight ($M_n$), and PDI (which equals to $M_w/M_n$) were calculated with the Cirrus™ GPC software (Agilent Technologies, USA).

### Reporting summary

Further information on research design is available in the Nature Portfolio Reporting Summary linked to this article.

## Data availability

Databases used in the study are: BRENDA database (www.brenda-enzymes.org/), eQuilibrator (https://equilibrator.weizmann.ac.il/). Data supporting the findings of this study are available within the article and its Supplementary Information. The source data underlying Fig. 3c, Fig. 4, Fig. 5, Supplementary Fig. 2, Supplementary Figs. 3a-3d, and Supplementary Figs. 4-14 are provided as a Source Data file. Source data are provided with this paper.

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

## Acknowledgements

This work was supported by the National Key R&D Program of China (grant numbers 2023YFA0914000 and 2021YFA0910601, to C.Y.), the National Natural Science Foundation of China (grant number 32001027, to X.W.; grant numbers 32271473 and 32022044, to C.Y.), and the Tianjin Synthetic Biotechnology Innovation Capacity Improvement Project (grant numbers TSBICIP-KJGG-003 and TSBICIP-CXRC-063, to C.Y.).

## Author contributions

C.Y. conceived the in vitro synthetic enzymatic pathway. C.Y. and X.W. designed the experiments and analyzed the data. X.W., L.X., and F.L. constructed the plasmids. X.W., Q. Li, C.H., and X.N. prepared the enzymes and tested the enzyme activities. X.W. and X.Y. constructed the computational models and performed related analyses. X.W. performed the experiments to produce PHB from maltodextrin. Q. Liu and Y.W. performed molecular weight analyses. X.W. and C.Y. wrote the initial version of the manuscript. C.Y., X.W., X.Y., X.C., T.C., Y.Z., Z.Z., and Y.-H.P. J.Z. contributed to the editing of the manuscript. All authors have given approval to the final version of the manuscript.

## Competing interests

The authors declare no competing interests.
