## [Peer Review File · Nature Communications]

ATP-free in vitro biotransformation of starch-derived maltodextrin into poly-3-hydroxybutyrate via acetyl-CoAEditorial Note: Parts of this Peer Review File have been redacted as indicated to remove third-party material where no permission to publish could be obtained.

REVIEWER COMMENTS

Reviewer #1 (Remarks to the Author):

This work and the authors are to be commended for showing that pathway simulations reflecting both stoichiometries and, to some extent, dynamics can contribute to the design and troubleshooting of pathways involving significant molecular rearrangements. In particular, the successful use of the transketolase and transenolase enzymes of the non-oxidative pentose pathway will be encouraging to other investigators. While this work indicates significant progress relative to a prior publication from the same group, it does not appear to be a major advance relative to that manuscript

It is troubling that the authors repeatedly refer to the production of the PHB polymer from starch when in fact the pathway results begin with a debranched maltodextrin which is at least two enzymatic steps and a separation step away from crude starch. Also, significant aspects of the PHB production kinetics are not recognized and discussed. In particular, while the initial production rate is impressive, it rapidly declines. Also, a secondary incubation is required for full processing of the initial substrate (and the kinetics of this phase are not shown). Finally, it is troubling that the required enzymatic activity of the final enzyme, PhaC, is approximately 50 times what would appear to be required to support the PHB production rate. While the authors claim to have demonstrated "scale-up" by doubling the initial maltodextrin concentration, they also have to double the enzyme concentrations rather than showing that more substrate can be consumed over a longer period of time with the same enzyme concentrations. All of this points to apparently unrecognized rate limitations that intensify as the reaction proceeds. This reviewer suggests that these issues need to be characterized and mitigated before this technology platform can provide a sufficiently important advance for acceptance by this important journal.

Reviewer #2 (Remarks to the Author):

This is a very nice paper describing how to establish a cell free system that does not require regeneration of ATP for conversion of starch to acetyl-CoA derived chemicals, here illustrated by PHB production. By using starch it is possible to phosphorylate glucose (in the first position) with free phosphate, and by relying on the pentose phosphate pathway together with the phosphoketolase reaction it is possible to generate acetyl-CoA without ATP requirement. Furthermore, through these pathways there are generated NADPH required for conversion of acetyl-CoA to PHB. The approach is elegant and also very well executed. What I particular like is that the authors combine the experimental execution of the strategy with mathematical modeling design for identification of the right enzyme concentrations. The paper is also very well written and I do not have any specific comments for improvement. Only overall comment I have is that the discussion feels long and could maybe benefit from shortening to make the key messages come out more clear.

Reviewer #3 (Remarks to the Author):

This work demonstrates a new system of metabolic reactions to create the bioplastic PHB from starch. It carefully explores two similar but distinct metabolic routes that utilize the bifunctional activity of the enzyme phosphoketolase (PKL) to produce the key intermediate acetyl-CoA as well as reducing equivalents of NADPH to produce PHB. This work is a significant to the cell-free metabolic engineering and in vitro enzyme cascade communities and provides an exciting advance as an alternative pathway to acetyl-CoA that does not utilize ATP (helpfully summarize in Table 1 by the authors). They also helpfully describe their work in the context of recent valuable studies by the Bowie lab (Opgenorth et al 2016). In general, the experiments support the conclusions, the methodology is sound, and methods have been thoughtfully and clearly described though there are a few minor comments need to be addressed.

Page 8 – the amount of PhaA loaded uses inconsistent units. The text reads: “Each enzyme was loaded at a final concentration of 1 U/mL, except for PhaA, which was loaded at 1 mg/mL due to its low specific activity” Please provide both the enzyme concentration (1 mg/mL) and enzyme unit loading (XX U/mL) or explain why this inconsistent description was chosen.

Page 27. The Enzymatic activity assays section of the methods appears incomplete with no description of how α GP, PGM, PGI, PhaC, TAL, TK, RPI, RPE, Tim, ALD, and FBP activities were measured. ‘

Page 9 and legend of Supplementary Figure 2– the use of “esterified” to describe the gas chromatography method is misleading (esterification refers to creating an ester from an alcohol and acid). Based on ref 28, the authors appear to have measured “methyl esters produced from acidic methanolysis”. Please clarify this word choice.

For clarity of Figure 4, Supplementary Figure 3, and Supplementary Figure 5, the authors should add to the figure panels the label “1X enzyme loading” and “5X enzyme loading” as appropriate

I appreciate the authors inclusion and analysis of Supplementary Figure 4 testing various levels of supplemental NADP+. CoA and TPP are also expensive cofactors supplemented to the In vitro biotransformation system. It would increase the strength of the manuscript to demonstrate that CoA and TPP are supplemented at optimal levels?

This is a fine distinction but can the authors note in the text that the optimization efforts described in Supplementary Figure 6 and 7 are in silico rather than experimental optimization? Thus page 12: “Guided by the optimization results, a subsequent trial of the starch-to-PHB experiment was carried out using the predicted optimal enzyme concentrations which are circled in Supplementary Fig. 7 and summarized in Supplementary Table 6” could read “Guided by the in silico optimization results, a subsequent trial of the starch-to-PHB experiment was carried out using the predicted optimal enzyme concentrations which are circled in Supplementary Fig. 7 and summarized in Supplementary Table 6”

To better put these results in context, it would be helpful to understand the physical enzyme loading parameters. Can the authors please specify in a supplementary table the mg/mL and/or molar concentrations of enzyme that corresponds to the loading in each figure? I appreciate that there is difficulty in precision here since Supplementary Figure 1

shows that TK, TIM, and G6PDH may have contaminant protein from the purification process but approximations (where the uncertainty is noted or estimated based on band densitometry) would still be useful.

The in silico model of is an exciting and useful body of work. Why did the authors choose to show only two rounds of optimization? For instance, in Supplementary Figure 7, the model predicts that the optimal levels of TAL and RPE are low (analogous to PKL). It would follow that running additional rounds to determine if a higher levels of TAL, RPE, and PKL are optimal? Since the authors highlight in the introduction the engineering flexibility of In vitro biotransformation (ivBT), I suggest they demonstrate that by testing more than three enzyme loadings (1 U/mL, 5 U/mL, after optimization; this would also provide more data points to assess the predictive capacity of the model.

I found that it was important to understand that Figure 4b and Figure 5a have similar rates and titers of PHD production (though with Fig 5a utilizing reduced enzyme loading). To enhance the readers' ability to interpret the results and compare that Figure 4b and Figure 5a visually, I suggest that the flow of the paper will not be altered if Fig 4 and 5 are combined into a single figure but ultimately trust the authors intuition on this point.

Is there a reason that the authors do not provide supplemental data (i.e. time course plots of maltodextrin consumed and PHB produced) for the data presented in Figure 5b and the claims in the text "Furthermore, to investigate the scale-up potential of our ivSEB, the starch concentration was doubled to 200 mM in a 0.5 L reactor with 0.4 L reaction solution. Accordingly, the concentrations of all enzymes (including the auxiliary enzymes), cofactors, and phosphate ions were doubled, leading to the production of 202.4 ± 2.1 mM (21.1 ± 0.2 g/L) PHB at 36 h with a slightly decreased molar yield (102.6%)." These plots would be useful to the reader.

Table 1 is very informative in putting these results in context. Thank you for providing such clear comparisons that highlight this excellent work.

Reply to Reviewers' comments point-by-point:

Reviewer #1 (Remarks to the Author):

This work and the authors are to be commended for showing that pathway simulations reflecting both stoichiometries and, to some extent, dynamics can contribute to the design and troubleshooting of pathways involving significant molecular rearrangements. In particular, the successful use of the transketolase and transenolase enzymes of the non-oxidative pentose pathway will be encouraging to other investigators. While this work indicates significant progress relative to a prior publication from the same group, it does not appear to be a major advance relative to that manuscript.

Response:

We sincerely appreciate your commendation and insightful evaluation of our manuscript. We appreciate the opportunity to highlight the advances of our manuscript.

Regarding your mention of a prior publication from the same group, we have carefully considered your feedback. We believe there might be some ambiguity regarding the referenced publication. However, we have two potential references in mind that might align with your comments.

Our first assumption is that you might be referring to another group's work (reference #28 of our manuscript), which shares some similarities in the ivSEB pathway for PHB production. In comparison, we emphasize significant improvements in the introduction and discussion of our study. Specifically, we have advanced the field by:

1. Eradicating the reliance on costly and unstable ATP by adopting cheaper starch as the substrate instead of glucose. This alteration allows the generation of phosphate sugars from starch in the presence of inorganic phosphate. Subsequently, within the downstream pathway, the substitution of PfkB, an ATP-regenerating enzyme described in reference #28, with FBP for P_i regeneration has been pivotal. The utilization of FBP in lieu of PfkB potentially contributes to an escalated PHB production rate through a more energy-favorable reaction.
2. Highlighting the importance of stoichiometric analysis in ivSEB pathways for accurate theoretical yield estimation and pathway design. This allowed us to identify discrepancies in the calculation of theoretical PHB yield in reference #28 and eliminate the use of NADP(H) purge valves in our ivSEB.
3. Streamlining the reaction process by simplifying PHB collection, as opposed to the multiple collection rounds required in reference #28, thereby enhancing operational efficiency.

4. Achieving a notably higher PHB molar yield of around 120% from 100 mM glucose equivalent of substrate after system optimization guided by the *in silico* model, compared to the approximately 85% yield reported in reference #28.

Our second assumption pertains to the possibility of referencing our previous publication (Wei, X. *et al.*, ChemCatChem, 2018, 10(24), 5597-5601). This ivSEB features regenerating three molecules of ATP from one glucose equivalent of starch in the absence of NAD(P) and CoA. While there might be shared enzymes between our prior ivSEB and the ivSEB of our current study, the focus of the studies and the product of the ivSEBs differ substantially, limiting direct comparison.

We greatly value your feedback which will undoubtedly contribute to the refinement and integrity of our manuscript. We hope our response addresses your concerns.

Action:

We have revised the first paragraph of our Discussion section (pages 17-18 in the clean version) to emphasize the importance of our work as follows:

“... Compared with the glucose-based ivSEB reported by Opgenorth *et al.*²⁸, our ivSEB presented several key enhancements. Firstly, our ivSEB exclusively used starch or maltodextrin as substrate to generate phosphate sugars with P_i, bypassing the need for costly and unstable ATP. Secondly, our ivSEB replaced PfkB-catalyzed ATP regeneration ($\Delta_r G^\circ = 16.9 \pm 1.3$ kJ/mol) in the glucose-based system with the energy-favorable FBP-catalyzed P_i regeneration ($\Delta_r G^\circ = -12.4 \pm 1.4$ kJ/mol), which may partly account for the much higher PHB production rate of our system (**Table 1**). Thirdly, our pathway eliminated the need for NAD(P)H purge valves, validated by the model-based stoichiometric analysis and quantitative experiments. Fourthly, our process achieved a notably enhanced PHB molar yield of around 120% from 100 mM glucose equivalents of substrate, compared to the approximately 85% yield reported in the glucose-based ivSEB. Lastly, our process simplified operations by requiring a single PhaC loading at the outset, thereby eliminating the need for multiple steps involving product removal and PhaC re-addition. Importantly, in contrast to the challenges associated with separating PHB from microbial cells through fermentation³⁸, our ivSEB facilitates the easy separation of insoluble PHB produced. While there may be room for further improvements, this study reveals the industrial potential of our designed ivSEB for the high-yield production of PHB from starch-derived substrates. Additionally, our designed pathway could be adapted to synthesize other acetyl-CoA-derived products, such as n-butanol, lipids, and isoprenoids by altering downstream enzymes.”

It is troubling that the authors repeatedly refer to the production of the PHB polymer from starch when in fact the pathway results begin with a debranched maltodextrin which is at least two enzymatic steps and a separation step away from crude starch.

Response:

We appreciate your keen observation regarding the distinction between starch and debranched maltodextrin in our manuscript.

Our choice to utilize maltodextrin for PHB synthesis stems from practical considerations. The

heightened solubility of maltodextrin, as opposed to crude starch, enables experimentation at higher substrate concentrations. Moreover, considering the insolubility of PHB, the utility of soluble maltodextrin proves advantageous over insoluble starch, significantly enhancing the efficiency of PHB separation.

Indeed, maltodextrin used in this study, derived from starch, undergoes additional enzymatic steps and a separation process before initiating the PHB polymer pathway. However, the production of maltodextrin from starch is a well-established and cost-effective industrial process. This technique, supported by mature industrial infrastructure, ensures a cost-efficient supply chain for low price of maltodextrin. Furthermore, separation step could be avoided, because the maltodextrin in the solution from the established industrial process of starch liquefaction can be utilized directly as the substrate for debranching, followed by PHB production.

We acknowledge your concerns and have attempted PHB production directly from various edible crude starch sources. This investigation could illustrate the versatility of our ivSEB approach for PHB production. However, we observed that the utilization of these crude starch sources as substrates resulted in slower PHB production rates and reduced PHB titers compared to maltodextrin, which may attribute to the low solubility of crude starch. Additionally, the inherent stickiness of starch solutions posed handling challenges, unlike maltodextrin.

We have incorporated this clarification regarding the choice of maltodextrin over starch and the attempted use of edible starch into our manuscript. Your insights are invaluable in refining the accuracy and comprehensiveness of our work.

Action:

To ensure clarity, we have made pivotal additions and alterations to the manuscript based on your valuable insights.

1. We have included **Supplementary Fig. 11**, detailing our endeavors to produce PHB from four distinct edible starch sources (corn, yam, cassava, and wheat). We have also added a new paragraph in our Results (page 14 lines 1-12 in the clean version) describing this experiment and the reason why we preferred maltodextrin over starch:

“Using the optimized enzyme and cofactor concentrations, we further evaluated the performance of our ivSEB in converting crude starch to PHB. Edible crude starch from four different sources – corn, yam, cassava, and wheat (**Supplementary Fig. 11a**) – were pre-treated by IA and added to the reaction system at a final concentration of approximately 80 – 100 mM glucose equivalent. After a 10-hour reaction period, production of PHB from these starches ranged from around 26.2 to 35.1 mM (2.7 – 3.7 g/L), correlating to molar yields of 95.7 – 105.9% based on the consumed starch (**Supplementary Fig. 11b** and **11c**). These results affirmed the feasibility of our ivSEB for the conversion of crude starch to PHB. However, considering the poor solubility of starch – a potential factor contributing to lower PHB titers compared to soluble maltodextrin – and recognizing that soluble maltodextrin facilitates the separation of insoluble PHB, we intended to persist in employing maltodextrin as the substrate for subsequent investigations, with a specific focus on achieving complete substrate utilization.”

[FIGURE PANEL REDACTED]

Supplementary Fig. 11 | Production of PHB from edible crude starch. a, Four types of edible crude starch from corn, yam, cassava, and wheat, respectively, were used for the production of PHB. **b**, Amount of residual starch in the reaction solutions after reaction for 0, 8, and 10 h. **c**, Amount of PHB in the reaction solutions after reaction for 0, 8, and 10 h. Experimental results were displayed as mean \pm standard deviation (SD) of three replicates.

2. In the Discussion section (page 19 lines 12-14 in the clean version), we have introduced a new proposal for future research involving the utilization of maltodextrin obtained through the well-established industrial process of starch liquefaction. Specifically, we suggest exploring the potential of this maltodextrin as a direct substrate for *in vitro* debranching and subsequent PHB production:

“Thirdly, the maltodextrin derived from the established industrial process of starch liquefaction may serve as a direct substrate for *in vitro* debranching and PHB production in the future.”

3. Additionally, we have revised the manuscript, substituting "starch" with "maltodextrin" wherever necessary to ensure clarity and avoid misunderstandings.

Also, significant aspects of the PHB production kinetics are not recognized and discussed. In particular, while the initial production rate is impressive, it rapidly declines.

Response:

We greatly appreciate your astute observation. The observed decline in the production rate over time aligns with the typical kinetics of enzymatic reactions, where the gradual depletion of substrate commonly leads to a reduction in the reaction rate. To address this, we conducted a new experiment, introducing additional 100 mM maltodextrin at 4 h. Remarkably, this adjustment yielded a reaction rate during the 4 – 8 h period that closely mirrored that of the 0 – 4 h interval, as depicted in **Fig. 5b** of the revised manuscript.

To provide a more comprehensive understanding of the maltodextrin and PHB kinetics, we have implemented the following actions:

Action:

1. We have revised **Fig. 5**, specifically introducing a new experiment depicted in **Fig. 5b**, which illustrates a fed-batch addition of a total of 200 mM maltodextrin.

Fig. 5 | One-pot, two-step production of PHB aiming at achieving complete maltodextrin utilization.

a, Production of PHB from 100 mM glucose equivalent of maltodextrin. Initial reaction conditions were the same as those of Fig. 4d (refer to Methods for details), except that PhaC was added at 20 U/mL as suggested by Model 2 (see Supplementary Fig. 12). At 8 h, 4GT, PPGK, and polyphosphate were added to the system, reaching final concentrations of 0.2 U/mL (approximately 0.67 mg/mL), 2 U/mL (approximately 0.02 mg/mL), and 20 mM, respectively. **b**, Production of PHB from 200 mM glucose equivalent of maltodextrin using a fed-batch substrate addition strategy. Initial reaction conditions aligned with those of Fig. 5a. At 4 h, another 100 mM maltodextrin and 20 U/mL PhaC was added. At 12 h, 4GT,

PPGK, and polyphosphate were added to the system at the same concentrations as in Fig. 5a. c, Production of PHB from 200 mM glucose equivalent of maltodextrin with a single substrate addition. Enzymes, NADP⁺, CoA, TPP, and phosphate ion concentrations were doubled compared to those in Fig. 5a and 5b. At 8 h, 4GT, PPGK, and polyphosphate were added to the system at doubled concentrations as in Fig. 5a and 5b. Experimental results were displayed as mean ± standard deviation (SD) of three replicates.

2. Descriptions detailing the variations in maltodextrin and PHB concentrations over time of each figure have been incorporated into the Results section, ensuring a more detailed presentation of the experimental results:

“The reaction exhibited rapid kinetics in the first four hours, followed by a deceleration, reaching equilibrium at approximately 8 h, resulting in the production of 74.9 ± 0.5 mM (7.8 ± 0.1 g/L) PHB from the consumption of 59.7 ± 1.6 mM maltodextrin at a production rate of 9.4 mM/h (approximately 1.0 g/L/h) (**Fig. 4d**).” (the final six lines on page 13 in the clean version)

“In pursuit of achieving the full conversion of 100 mM maltodextrin into PHB, a one-pot two-step reaction was conducted (**Fig. 5a**). The first step, spanning from 0 to 8 h, replicated a previous experiment depicted in **Fig. 4d**, Subsequently, the second step commenced at 8 h when 4GT, PPGK, and polyphosphate were added. ... At these optimized enzyme concentrations, the reaction exhibited rapid progress during 8 – 12 h, followed by a slowing down. The near-complete maltodextrin consumption and the highest PHB titer were both achieved at 24 h, yielding ...” (page 15 lines 3 – 13 in the clean version)

“... we then implemented a fed-batch substrate addition strategy, adding the same amount of maltodextrin at 0 h and 4 h to achieve a final concentration of 200 mM. In contrast to **Fig. 5a**, where the reaction slowed down after 4 h without additional substrate, this time, the reaction rate during 4 – 8 h was similar to that of 0 – 4 h, suggesting the potential resolution of the substrate inhibition issue. At 12 h, 4GT and PPGK were added at the previously optimized concentrations. The reaction continued until 24 h, reaching an equilibrium with the production of 202.8 ± 4.4 mM (21.1 ± 0.5 g/L) PHB from the consumption of 185.8 ± 0.5 mM maltodextrin, corresponding to a PHB molar yield of 109.1% and an overall reaction rate of 8.5 mM/h (approximately 0.9 g/L/h) (**Fig. 5b**). To simplify the handling process, doubling the concentrations of all enzymes (including auxiliary enzymes), cofactors, and phosphate ions, along with a single addition of 200 mM maltodextrin at the beginning, resulted in an enhanced reaction rate during the first 4 hours (**Fig. 5c**). Followed by the addition of 4GT and PPGK at 8 h, the reaction proceeded at a relatively constant rate till 24 h, when 208.3 ± 10.0 mM (21.7 ± 1.0 g/L) PHB was produced from the consumption of 186.1 ± 2.6 mM maltodextrin. This strategy yielded similar results to those of the fed-batch experiment of **Fig. 5b** with a slightly enhanced molar yield of 111.9% and a slightly increased overall reaction rate of 8.7 mM/h (approximately 0.9 g/L/h) (**Fig. 5c**).” (page 16 lines 2 – 18 in the clean version)

Also, a secondary incubation is required for full processing of the initial substrate (and the kinetics of this phase are not shown).

Response:

Thank you for your valuable comment regarding the secondary incubation necessary for complete substrate processing. The rationale of this secondary incubation involving 4- α -glucanotransferase (4GT) and polyphosphate glucokinase (PPGK) has been meticulously illustrated in references #34 and #35 in our revised manuscript. To elaborate, 4GT catalyzes the transglycosylation among maltooligosaccharides, converting including maltose and maltotriose, yielding into longer maltodextrin that α -glucan phosphorylase (α GP) can utilize. This transglycosylation process also generates glucose, which can be converted to glucose 6-phosphate (G6P) in the presence of polyphosphate by PPGK. Importantly, the addition of these two enzymes, especially 4GT, must be sequenced as a second stage, initiated after the 17-enzyme ivSEB reaction attains equilibrium. Premature supplementation of 4GT may lead to reduced maltodextrin utilization efficiency due to its side reaction involving cyclization of long-chain starch/maltodextrin (supported by references #32 and #36 in our revised manuscript). The substrate for PPGK, which is glucose, is absent at the beginning of the reaction, and hence PPGK should be added in the second step alongside 4GT.

Regarding the kinetics of this second step of reaction, a similar concern was raised by Reviewer #3. In response, the revised manuscript now includes experiments with enhanced sampling frequency, resulting in a new **Fig. 5** (displayed on the page 5 of this point-by-point response). This figure presents a detailed time course of maltodextrin consumption and PHB production during both reaction stages. Furthermore, we have conducted experimental-based optimization for the concentrations of 4GT and PPGK (**Supplementary Fig. 13** in the revised manuscript), leading to an improved reaction rate compared to previous results. This enhancement directly stems from your insightful comment, significantly contributing to better outcomes in our study.

Action:

In response to your request, we have revised the related Results section (page 14, the first paragraph of the subsection “The complete conversion of maltodextrin to PHB” in the clean version) as follows:

“During PHB production, maltodextrin underwent gradual shortening to maltose, posing difficulties in its further utilization by α GP¹⁴. This incomplete utilization of maltodextrin within our designed 17-enzyme ivSEB prompted the incorporation of a second reaction step involving two auxiliary enzymes, 4- α -glucanotransferase (4GT) and polyphosphate glucokinase (PPGK). 4GT catalyzes the transglycosylation among maltooligosaccharides, converting maltose into longer maltodextrin that α GP can utilize³⁴. This transglycosylation process also generates glucose, which can be converted to G6P in the presence of polyphosphate by PPGK³⁵. Importantly, the addition of these two enzymes, especially 4GT, must be sequenced as a second stage, initiated after the 17-enzyme ivSEB reaction attains equilibrium. Premature supplementation of 4GT may lead to reduced maltodextrin utilization efficiency due to its side reaction involving cyclization of long-chain starch/maltodextrin ^{32,36}.”

We have also incorporated the necessary data into the revised manuscript:

1. **Supplementary Fig. 13** details the outcomes after the addition of 4GT and PPGK at the 8th hour, using 100 mM IA-treated maltodextrin as the substrate, illustrating the optimization of 4GT and PPGK concentrations.

Supplementary Fig. 13 | Experimental optimization of the loading amounts of 4GT and PPGK. 100 mM IA-debranched maltodextrin was used as the substrate. The reaction was initiated under the same conditions as that in Fig. 4d, except that the concentration of PhaC was raised to 20 U/mL. At 8 h, 4GT, and PPGK were added to the reaction mixture at either 1-fold concentrations (0.1 U/mL or approximately 0.33 mg/mL 4GT, 1.0 U/mL or approximately 0.01 mg/mL PPGK) or 2-fold concentrations (0.2 U/mL or approximately 0.67 mg/mL 4GT, 2.0 U/mL or approximately 0.01 mg/mL PPGK) together with 20 mM sodium hexametaphosphate for further PHB production.

2. **Fig. 5a** summarizes results from experiments using 100 mM IA-treated maltodextrin as the substrate, displaying a detailed time course of maltodextrin consumption and PHB production across both reaction stages.
3. **Fig. 5b** summarizes results from experiments using 200 mM IA-treated maltodextrin as the substrate, displaying a detailed time course of maltodextrin consumption and PHB production across both reaction stages. Except that PhaC was added at a higher concentration, as suggested by *in silico* simulation optimization, concentrations of other enzymes and cofactors remained the same as in Fig. 5a. Maltodextrin was added in a fed-batch mode.
4. **Fig. 5c** summarizes results from experiments using 200 mM IA-treated maltodextrin as the substrate, displaying a detailed time course of maltodextrin consumption and PHB production across both reaction stages. Enzyme and cofactor concentrations were doubled as in Fig. 5b. Maltodextrin was added by a single addition at 0 h.

(**Fig. 5** is displayed on the page 5 of this point-by-point response).

Finally, it is troubling that the required enzymatic activity of the final enzyme, PhaC, is approximately 50 times what would appear to be required to support the PHB production rate.

Response:

Thank you for this valuable feedback on our manuscript regarding the enzymatic activity of PhaC in the production of PHB. We appreciate the opportunity to address your concern.

In our initial experiments, we examined PhaC at concentrations of 1 U/mL and 5 U/mL, observing that only the higher concentration (5 U/mL) facilitated both a rapid PHB production rate and a notably higher PHB yield, as indicated in Fig. 4 (now relocated as **Fig. S3** in our revised manuscript).

Informed by the subsequent *in silico* optimization, we incrementally increased the PhaC concentration from 5 to 8 U/mL (a less-than-two-fold increase) to augment the product titer. This adjustment resulted in the near-complete conversion of approximately 60 mM glucose equivalent of maltodextrin into PHB. Despite the increased PhaC requirement after optimization, the total enzyme loading concentration was decreased by about 40% (see **Supplementary Table 6**).

Our decision to further elevate the PhaC concentration was driven by the need to enhance PHB production by utilizing more substrate. Consequently, a PhaC concentration of 20 U/mL was employed for the near-complete conversion of 100 mM substrate, as suggested by our kinetic model (depicted in **Supplementary Fig.12** in the revised manuscript). For the near-complete conversion of 200 mM substrate, we investigated the used of 40 U/mL PhaC, which was a simple doubling of the PhaC concentration compared to the 100 mM substrate scenario. We believe it is essential not to directly compare these PhaC concentrations with the earlier ones (i.e., 1 U/mL, 5 U/mL, and 8 U/mL for the complete conversion of around 60 mM substrate) without considering the concurrent increase in substrate concentration.

Acknowledging concerns about the escalating PhaC concentration and its impact on production costs, we recognize the need to address this aspect in our future studies. We discussed exploring machine learning algorithms and implementing strategies such as enzyme engineering and gene mining to enhance the activity and stability of PhaC, thereby reducing the required amount of PhaC.

Your insightful comments will be invaluable in guiding our future research endeavors, and we appreciate the opportunity to incorporate them into our ongoing work.

Action:

In our revised manuscript, specifically in the Discussion section spanning page 18 (the last line) and page 19 (lines 1-12) of the clean version of our revised manuscript, we have incorporated a discussion addressing the challenges posed by the high concentrations of enzymes on production costs. Additionally, we have proposed potential solutions, including the refinement of our kinetic model, the application of machine learning techniques to optimize enzyme concentrations within our ivSEB, and the exploration of enzyme engineering and gene mining approaches aimed at enhancing enzyme activity and stability. This part of the Discussion is shown as below:

“Firstly, despite the effective application of our semi-quantitative model in facilitating two-round *in silico* optimization of enzyme concentrations in this study, it is acknowledged that constructing precise kinetic models for complex ivSEBs presents challenges. Diverse factors, including mass transport, protein-protein interactions, substrate/product inhibition, and enzyme/cofactor degradation, can impact enzymatic reaction rates^{13,41,43}. Recent research has successfully employed machine learning to optimize enzyme concentrations in ivSEBs⁴⁴, and this method holds promise for optimizing our system based on a substantial amount of precise data. Secondly, reducing enzyme loading concentrations is crucial to decrease the enzyme cost of ivBT. Beyond optimizing enzyme concentrations, enhancing the activity and stability of enzymes involved in our ivSEB by protein engineering (aided by machine learning algorithms⁴⁵) and gene mining⁴⁶ is another important direction. This is particularly pertinent for PhaC, given its binding to the produced PHB^{28,47}, and for α GP, PhaA, and PhaB, which exhibited relatively low specific activity values and therefore required larger loading amounts in our study.”

While the authors claim to have demonstrated "scale-up" by doubling the initial maltodextrin concentration, they also have to double the enzyme concentrations rather than showing that more substrate can be consumed over a longer period of time with the same enzyme concentrations.

Response:

Thank you for the valuable comment and suggestion. We appreciate the opportunity to clarify our approach regarding the scale-up demonstration in response to your query.

In our initial manuscript submission, we aimed to assess the scale-up potential of our ivSEB by doubling the starch concentration to 200 mM. This adjustment prompted a corresponding doubling of concentrations of all enzymes, cofactors, and phosphate ions, as depicted in **Fig. 5c** in our revised manuscript.

While we initially assessed more substrate consumption by doubling enzyme concentrations, your insightful comment regarding examining substrate consumption over an extended period at consistent enzyme concentrations prompted us to conduct two additional experiments. In these experiments, we retained the concentrations of enzymes, cofactors and phosphate ions while doubling the substrate concentration to 200 mM. In the first experiment, 200 mM maltodextrin was supplemented as a single addition at 0 h. Surprisingly, this resulted in a noticeable decline in both reaction rate and PHB yield, as illustrated in the **Supplementary Fig. 14** in our revised manuscript. A potential cause for this decline might be substrate inhibition. For further investigation, our second experiment took a fed-batch strategy of maltodextrin addition (**Fig. 5b**). 100 mM maltodextrin was added at 0h, followed by the addition of another 100 mM maltodextrin at 4 h, and the addition of 4GT and PPGK at 12 h. This approach resulted in a system performance at 24 h similar to our previous experiment in **Fig. 5c**, conducted with doubled enzyme concentrations and a single addition of 200 mM maltodextrin at 0 h.

We genuinely appreciate your guidance, which has been instrumental in prompting these additional investigations and contributing to the depth of our study.

Action:

We have revised our Results section by adding a new subsection titled "The complete conversion of maltodextrin to PHB". In the last paragraph of this subsection, we described the results of **Fig. 5** and **Supplementary Fig. 14**, both of which are related to our scale-up investigation:

"Furthermore, to investigate the scalability of our ivSEB, we doubled the maltodextrin concentration to 200 mM. Initially maintaining the concentrations of enzymes, cofactors, and phosphate ions as in **Fig. 5a**, a single addition of 200 mM maltodextrin at 0 h resulted in a sluggish reaction rate and a low PHB product titer during the 24-hour reaction period (**Supplementary Fig. 14**). Suspecting substrate inhibition as the cause for the low reaction rate and PHB titer, we then implemented a fed-batch substrate addition strategy, adding the same amount of maltodextrin at 0 h and 4 h to achieve a final concentration of 200 mM. In contrast to **Fig. 5a**, where the reaction slowed down after 4 h without additional substrate, this time, the reaction rate during 4 – 8 h was similar to that of 0 – 4 h, suggesting the potential resolution of the substrate inhibition issue. At 12 h, 4GT and PPGK were added at the previously optimized

concentrations. The reaction continued until 24 h, reaching an equilibrium with the production of 202.8 ± 4.4 mM (21.1 ± 0.5 g/L) PHB from the consumption of 185.8 ± 0.5 mM maltodextrin, corresponding to a PHB molar yield of 109.1% and an overall reaction rate of 8.5 mM/h (approximately 0.9 g/L/h) (**Fig. 5b**). To simplify the handling process, doubling the concentrations of all enzymes (including auxiliary enzymes), cofactors, and phosphate ions, along with a single addition of 200 mM maltodextrin at the beginning, resulted in an enhanced reaction rate during the first 4 hours (**Fig. 5c**). Followed by the addition of 4GT and PPGK at 8 h, the reaction proceeded at a relatively constant rate till 24 h, when 208.3 ± 10.0 mM (21.7 ± 1.0 g/L) PHB was produced from the consumption of 186.1 ± 2.6 mM maltodextrin. This strategy yielded similar results to those of the fed-batch experiment of **Fig. 5b** with a slightly enhanced molar yield of 111.9% and a slightly increased overall reaction rate of 8.7 mM/h (approximately 0.9 g/L/h) (**Fig. 5c**). Compared with previously reported PHB-producing ivSEBs, our system not only provided a straightforward operational process for producing PHB from starch, but also achieved the near-theoretical product yield, the highest product titer, and the fastest reaction rate (**Table 1**), indicating that this study laid a solid foundation for scaling up the biosynthesis of acetyl-CoA-derived products from starch in an industrial setting.”

Supplementary Fig. 14 | Production of PHB from 200 mM glucose equivalent of maltodextrin. Concentrations of enzymes, cofactors, and phosphate ions remained the same as those used for 100 mM substrate in **Fig. 5a**. Experimental results were displayed as mean \pm standard deviation (SD) of three replicates.

(**Fig. 5** is displayed on the page 5 of this point-by-point response).

All of this points to apparently unrecognized rate limitations that intensify as the reaction proceeds. This reviewer suggests that these issues need to be characterized and mitigated before this technology platform can provide a sufficiently important advance for acceptance by this important journal.

Response:

We sincerely appreciate your thoughtful comment and the insights you've provided. Building upon our previous Responses and Actions, we have made revisions to **Fig. 5** to address the observed

slowdown in the reaction rate after 4 hours, as shown in **Fig. 5a**. It is now evident that this deceleration may be attributed to the maltodextrin chain becoming too short for further utilization by α GP. To mitigate this, we have introduced two strategies: the addition of enzymes 4GT and PPGK for the comprehensive utilization of short-chain maltodextrin (as illustrated in **Fig. 5a**), or alternatively, by adding more maltodextrin at 4 h (depicted in **Fig. 5b**). We believe these modifications provide meaningful insights into the reaction kinetics, addressing the concerns you raised regarding rate limitations. Your guidance is invaluable, and we are committed to refining our approach to meet the rigorous standards of this esteemed journal.

Reviewer #2 (Remarks to the Author):

This is a very nice paper describing how to establish a cell free system that does not require regeneration of ATP for conversion of starch to acetyl-CoA derived chemicals, here illustrated by PHB production. By using starch it is possible to phosphorylate glucose (in the first position) with free phosphate, and by relying on the pentose phosphate pathway together with the phosphoketolase reaction it is possible to generate acetyl-CoA without ATP requirement. Furthermore, through these pathways there are generated NADPH required for conversion of acetyl-CoA to PHB. The approach is elegant and also very well executed. What I particular like is that the authors combine the experimental execution of the strategy with mathematical modeling design for identification of the right enzyme concentrations. The paper is also very well written and I do not have any specific comments for improvement. Only overall comment I have is that the discussion feels long and could maybe benefit from shortening to make the key messages come out more clear.

Response:

We sincerely appreciate your positive evaluation of our study and the insightful comments provided. We have taken into consideration your point raised and revised our Discussion section.

Action:

In response to your feedback, we have diligently revised the Discussion section with a primary emphasis on conciseness while ensuring the retention of essential content and key messages. The original Discussion section (approximately 1230 words) has been streamlined to around 920 words. The revised version comprises four paragraphs:

1. The first paragraph focused on the advantageous of our ivSEB in comparison to previous studies.
2. The second paragraph discussed the importance of stoichiometric analysis, which is a pivotal aspect of our research that holds the potential to benefit future studies on complex ivSEBs.
3. The third paragraph proposed future works related to our study.
4. The final paragraph summarized the significance and implications of our study.

Reviewer #3 (Remarks to the Author):

This work demonstrates a new system of metabolic reactions to create the bioplastic PHB from starch. It carefully explores two similar but distinct metabolic routes that utilize the bifunctional activity of the enzyme phosphoketolase (PKL) to produce the key intermediate acetyl-CoA as well as reducing equivalents of NADPH to produce PHB. This work is a significant to the cell-free metabolic engineering and in vitro enzyme cascade communities and provides an exciting advance as an alternative pathway to acetyl-CoA that does not utilize ATP (helpfully summarize in Table 1 by the authors). They also helpfully describe their work in the context of recent valuable studies by the Bowie lab (Opgenorth et al 2016). In general, the experiments support the conclusions, the methodology is sound, and methods have been thoughtfully and clearly described though there are a few minor comments need to be addressed.

Response:

We greatly appreciate your positive evaluations. We have tried our best to provide detailed responses below, addressing each point raised.

Page 8 – the amount of PhaA loaded uses inconsistent units. The text reads: “Each enzyme was loaded at a final concentration of 1 U/mL, except for PhaA, which was loaded at 1 mg/mL due to its low specific activity” Please provide both the enzyme concentration (1 mg/mL) and enzyme unit loading (XX U/mL) or explain why this inconsistent description was chosen.

Response:

We appreciate your highlight of this inconsistency. As outlined in **Supplementary Table 1**, the specific activity of PhaA is 0.78 mU/mg. Hence, a concentration of 1 mg/mL of PhaA corresponds to approximately 0.78 mU/mL.

Our decision to use 1 mg/mL of PhaA was based on its low specific activity, rendering a concentration of 1 U/mL unfeasible due to the resulting mass concentration exceeding 1000 mg/mL. Therefore, for clarity, we opted to represent PhaA using mass concentration rather than enzyme unit concentration. The choice of 1 mg/mL or 1 U/mL for the proof-of-concept experiment was arbitrary; variations like starting with 0.5 or 2 can be equally effective, as the values will undergo optimization in subsequent steps.

Action:

To address this, we have included the value of 0.78 mU/mL alongside the 1 mg/mL PhaA concentration in the text for clarity and completeness.

We have also summarized the enzyme concentrations used for our experiments in both U/mL and mg/mL in **Supplementary Table 6** as well as the Source Data.

Page 27. The Enzymatic activity assays section of the methods appears incomplete with no description of how α GP, PGM, PGI, PhaC, TAL, TK, RPI, RPE, Tim, ALD, and FBP activities were measured. ‘

Response:

We appreciate your thorough examination of our Enzymatic Activity Assays section and acknowledge the absence of direct measurement descriptions for certain enzymes in our manuscript (α GP, PGM, PGI, PhaC, TAL, TK, RPI, RPE, TIM, ALD, and FBP). However, these enzymes have been characterized carefully in previous publications (for example, Wei, X. *et al.*, ChemCatChem, 2018, 10(24), 5597-5601). To address your concerns, we opted to reference established specific activity values from reputable literature sources. These references, detailing the methodology for activity assays, have been meticulously compiled in **Supplementary Table 1** under the column labeled "Reference for specific activity."

Your attention to detail is invaluable, and your feedback guides us in fortifying the completeness and clarity of our Methods section. We are committed to ensuring that our methodologies are transparent and comprehensively detailed for the benefit of our readers.

Page 9 and legend of Supplementary Figure 2– the use of “esterified” to describe the gas chromatography method is misleading (esterification refers to creating an ester from an alcohol and acid). Based on ref 28, the authors appear to have measured “methyl esters produced from acidic methanolysis”. Please clarify this word choice.

Response:

Thank you for the discerning observation regarding the terminology in our figure legend describing the gas chromatogram analysis. Upon reevaluation, we acknowledge the need for clarity in the description. The term "esterified" might indeed misrepresent the gas chromatography methodology. We appreciate your meticulous attention to detail, and your guidance will significantly contribute to enhancing the accuracy and clarity of our manuscript.

Actions:

1. To provide a more precise description, we have revised the legend as follows:

Supplementary Fig. 2 | Gas chromatographic profiles of methyl esters from PHB standards. **a**, Gas chromatogram showing profiles of methyl esters resulting from acidic methanolysis of PHB standards (yielding methyl 3HB) and benzoic acid (yielding methyl benzoate). The latter was employed as an internal standard for PHB quantification. **b**, Standard curve of PHB plotted based on the gas chromatographic results.

2. We have revised the relevant content in the “Quantitative evaluation of PHB” subsection of “Results” (page 9 lines 15-16 in the clean version) as follows:

“PHB in the sample was methanolized by heating at 95 °C in the presence of methanol and sulfuric acid, and detected by gas chromatography, ...”

3. We have revised the relevant content in the “Quantification of PHB” subsection of “Methods” as follows:

“Then, 1 mL chloroform and 1 mL methanolysis reagent (prepared by mixing 85 mL methanol, 15 mL concentrated sulfuric acid, and 0.7 g benzoic acid) were added to each dried

pellet. The mixture was heated at 95 °C for 4 h in a screw-cap glass tube, followed by cooling to room temperature. Subsequently, 1 mL deionized water was added for extraction and phase separation.”

“The peak representing methyl 3HB in a sample was determined using a PHB standard (Sigma) that underwent methanolysis via the same method, and the peak area was compared with that of methyl benzoate (produced from acidic methanolysis of benzoic acid which was used as an internal standard) in each sample for quantification.”

For clarity of Figure 4, Supplementary Figure 3, and Supplementary Figure 5, the authors should add to the figure panels the label “1X enzyme loading” and “5X enzyme loading” as appropriate.

Response:

We sincerely appreciate your suggestion.

Action:

In response, we have incorporated the recommended labels into these figures (which are **Fig. 4**, **Supplementary Fig. 3**, and **Supplementary Fig. 7** in the revised manuscript) to enhance clarity and facilitate better comprehension for readers.

Fig. 4 | One-pot, one-step production of PHB from 100 mM maltodextrin by the designed ivSEB. Experimental data were represented as dots, while simulation results were shown as dotted lines, generated using Model 2 which had undergone two rounds of model fitting. The first round of model fitting utilized experimental data from **Fig. 4a** and **4b**. The second round of model fitting used experimental data from **Fig. 4a**, **4b**, and **4c**. **a**, PhaA was loaded at 1 mg/mL (0.78 mU/mL) while other enzymes were loaded at 1 U/mL. **b**, PhaA was loaded at 5 mg/mL (3.9 mU/mL) while other enzymes were loaded at 5 U/mL. **c**, Enzymes were loaded at the suggested concentrations after the first round of *in silico* optimization, as summarized in **Supplementary Table 6**. **d**,

Enzymes were loaded at the suggested concentrations after the second round of *in silico* optimization, as summarized in **Supplementary Table 6**. Results were displayed as mean \pm standard deviation (SD) of three replicates.

Supplementary Fig. 3 | Production of PHB from 100 mM maltodextrin by the designed ivSEB. a and b, Each enzyme was loaded at 1 U/mL except that PhaA was loaded at 1 mg/mL (i.e. 1-fold enzyme concentrations). **c, d and e,** Each enzyme was loaded at 5 U/mL except that PhaA was loaded at 5 mg/mL (i.e. 5-fold enzyme concentrations). **a and c,** Gas chromatographic profiles of methanolized PHB samples produced from 100 mM IA-treated maltodextrin. **b and d,** Concentration profiles of maltodextrin and PHB. Results were displayed as mean \pm standard deviation (SD) of three replicates. **e,** Photos showing the reaction samples of **Supplementary Fig. 3d**.

Supplementary Fig. 7 | Comparisons of results predicted by the kinetic models with the experimental data before and after the first round of model fitting. 100 mM glucose equivalent of IA-treated maltodextrin was used as substrate. **a,b**, Comparisons of simulation data obtained from the unfitted Model 0 with experimental results. **c,d**, Comparisons of simulation data obtained from Model 1 (which is the model after the first round of model fitting) with experimental results. For **a** and **c**, V_{\max} of each enzyme in the model was set as 1 mM/min (except that FPK had a V_{\max} of 0.12 mM/min), and each enzyme was loaded at 1 U/mL (except that PhaA was loaded at 1 mg/mL or 0.78 mU/mL) for the experiment. For **b** and **d**, V_{\max} of each enzyme in the model was set as 5 mM/min (except that FPK had a V_{\max} of 0.6 mM/min), and each enzyme was loaded at 5 U/mL (except that PhaA was loaded at 5 mg/mL or 3.9 mU/mL) for the experiment. Model was constructed as described in the Methods. Kinetic functions and parameters used for model construction refer to **Supplementary Table 2**, **Supplementary Table 3**, and **Supplementary Table 5**. Experimental results were displayed as mean \pm standard deviation (SD) of three replicates.

I appreciate the authors inclusion and analysis of Supplementary Figure 4 testing various levels of supplemental NADP⁺. CoA and TPP are also expensive cofactors supplemented to the In vitro biotransformation system. It would increase the strength of the manuscript to demonstrate that CoA and TPP are supplemented at optimal levels?

Response:

We highly value your insightful suggestion regarding CoA and TPP supplementation. As per your guidance, we conducted optimization experiments for both cofactors. Our investigations revealed that within the range of 0 to 2 mM, a concentration of 0.5 mM CoA yielded the fastest initial reaction rate and the highest final PHB yield, aligning with our previous usage. However, our optimization for TPP indicated that 2 mM proved more optimal than the previously employed 0.5 mM, exhibiting better performance in both initial reaction rate and final PHB yield. Considering that TPP is an

expensive cofactor and the fact that our initial trial with 0.5 mM TPP already yielded a relatively fast reaction rate, we deem 2 mM to be sufficient. Therefore, we did not investigate the impact of higher concentrations of TPP. Consequently, these optimized concentrations of CoA and TPP were employed in subsequent investigations, particularly in **Fig. 4d**, showcasing our most recent findings utilizing these optimized enzyme and coenzyme concentrations.

Action:

We have introduced a new section named “Effect of cofactor input on the production of PHB” in our Results, which details the results of NADP⁺, CoA, and TPP supplementation optimization. The optimization results of CoA and TPP were displayed in **Supplementary Fig. 5** and **Supplementary Fig. 6**, respectively.

Supplementary Fig. 5 | Effect of CoA input concentrations on the production of PHB. **a**, Effect of CoA input concentration on initial PHB production rate (defined as the amount of PHB produced within the first hour). **b**, Effect of CoA input concentration on PHB titer at 8 h. The reactions were carried out at 37 °C in 200 mM Tris-HCl buffer (pH 7.4) containing 10 mM MgCl₂, 0.5 mM MnCl₂, 10 µg/mL ampicillin, 5 µg/mL kanamycin, 10 mM sodium phosphate (pH 7.4), 2 mM NADP⁺, 0.5 mM TPP, 100 mM IA-debranched maltodextrin, and enzymes. Each enzyme was loaded at 5 U/mL except that PhaA was loaded at 5 mg/mL or 3.9 mU/mL. Additional CoA were supplemented to the samples at 0 h, with their concentrations in the reaction mixture indicated on the x axes. Relative initial PHB production rates and PHB titers at 8 h were calculated using the results obtained at 0.5 mM CoA as 100%. Results were displayed as mean ± standard deviation (SD) of three replicates.

Supplementary Fig. 6 | Effect of TPP input concentrations on the production of PHB. **a**, Effect of TPP input concentration on initial PHB production rate (defined as the amount of PHB produced within the first hour). **b**, Effect of TPP input concentration on PHB titer at 8 h. The reactions were carried out at 37 °C in 200 mM Tris-HCl buffer (pH 7.4) containing 10 mM MgCl₂, 0.5 mM MnCl₂, 10 µg/mL ampicillin, 5 µg/mL kanamycin, 10 mM sodium phosphate (pH 7.4), 2 mM NADP⁺, 0.5 mM CoA, 100 mM IA-debranched maltodextrin, and

enzymes. Each enzyme was loaded at 5 U/mL except that PhaA was loaded at 5 mg/mL or 3.9 mU/mL. Additional TPP were supplemented to the samples at 0 h, with their concentrations in the reaction mixture indicated on the x axes. Relative initial PHB production rates and PHB titers at 8 h were calculated using the results obtained at 2 mM TPP as 100%. Results were displayed as mean \pm standard deviation (SD) of three replicates.

Moreover, the manuscript's revised sections now include experiments conducted at these optimized cofactor concentrations:

1. **Fig. 4d** in the revised manuscript depicts the production of PHB from 100 mM substrate at optimized enzyme and coenzyme concentrations within the initial 0 to 8-hour timeframe without the addition of 4GT and PPGK.

(**Fig. 4d** is displayed on the page 15 of this point-by-point response).

2. **Supplementary Fig. 13** in the revised Supplementary Information illustrates the PHB production from 100 mM substrate at the optimized enzyme and coenzyme concentrations, focusing on the time course at 8 hours post the addition of the different concentrations of 4GT and PPGK.

Supplementary Fig. 13 | Experimental optimization of the loading amounts of 4GT and PPGK. 100 mM IA-debranched maltodextrin was used as the substrate. The reaction was initiated under the same conditions as that in **Fig. 4d**, except that the concentration of PhaC was raised to 20 U/mL. At 8 h, 4GT, and PPGK were added to the reaction mixture at either 1-fold concentrations (0.1 U/mL or approximately 0.33 mg/mL 4GT, 1.0 U/mL or approximately 0.01 mg/mL PPGK) or 2-fold concentrations (0.2 U/mL or approximately 0.67 mg/mL 4GT, 2.0 U/mL or approximately 0.01 mg/mL PPGK) together with 20 mM sodium hexametaphosphate for further PHB production.

3. **Fig. 5** in the revised manuscript illustrates the one-pot, two-step production of PHB aiming at the complete conversion of maltodextrin, displaying a detailed time course of maltodextrin consumption and PHB production across both reaction steps.

Fig. 5 | One-pot, two-step production of PHB aiming at achieving complete maltodextrin utilization.

a, Production of PHB from 100 mM glucose equivalent of maltodextrin. Initial reaction conditions were the same as those of Fig. 4d (refer to Methods for details), except that PhaC was added at 20 U/mL as suggested by Model 2 (see Supplementary Fig. 12). At 8 h, 4GT, PPGK, and polyphosphate were added to the system, reaching final concentrations of 0.2 U/mL (approximately 0.67 mg/mL), 2 U/mL (approximately 0.02 mg/mL), and 20 mM, respectively. **b**, Production of PHB from 200 mM glucose equivalent of maltodextrin using a fed-batch substrate addition strategy. Initial reaction conditions aligned with those of Fig. 5a. At 4 h, another 100 mM maltodextrin and 20 U/mL PhaC was added. At 12 h, 4GT, PPGK, and polyphosphate were added to the system at the same concentrations as in Fig. 5a. **c**, Production of PHB from 200 mM glucose equivalent of maltodextrin with a single substrate addition. Enzymes, NADP⁺, CoA, TPP, and phosphate ion concentrations were doubled compared to those in Fig. 5a and 5b. At 8 h, 4GT, PPGK, and polyphosphate were added to the system at doubled concentrations as

in Fig. 5a and 5b. Experimental results were displayed as mean \pm standard deviation (SD) of three replicates.

The Methods section has also been revised accordingly.

This is a fine distinction but can the authors note in the text that the optimization efforts described in Supplementary Figure 6 and 7 are *in silico* rather than experimental optimization? Thus page 12: “Guided by the optimization results, a subsequent trial of the starch-to-PHB experiment was carried out using the predicted optimal enzyme concentrations which are circled in Supplementary Fig. 7 and summarized in Supplementary Table 6” could read “Guided by the *in silico* optimization results, a subsequent trial of the starch-to-PHB experiment was carried out using the predicted optimal enzyme concentrations which are circled in Supplementary Fig. 7 and summarized in Supplementary Table 6”.

Response:

Thank you for this valuable suggestion. Your guidance has been instrumental in ensuring precision and clarity in delineating between experimental and *in silico* optimization efforts within the manuscript.

Action:

In response, we have revised the text as advised, incorporating the term "*in silico*" on page 12, line 16:

“Guided by the *in silico* optimization results derived using Model 1 (**Supplementary Fig. 8**), a subsequent trial of the maltodextrin-to-PHB experiment was carried out employing the predicted optimal enzyme concentrations (summarized in **Supplementary Table 6**).

Additionally, we have included the term "*in silico*" across all sections referring to optimization efforts utilizing the kinetic model.

To better put these results in context, it would be helpful to understand the physical enzyme loading parameters. Can the authors please specify in a supplementary table the mg/mL and/or molar concentrations of enzyme that corresponds to the loading in each figure? I appreciate that there is difficulty in precision here since Supplementary Figure 1 shows that TK, TIM, and G6PDH may have contaminant protein from the purification process but approximations (where the uncertainty is noted or estimated based on band densitometry) would still be useful.

Response:

Thank you for the insightful suggestion. To enhance the contextual understanding of our results, we have incorporated enzyme loading parameters in both U/mL and mg/mL in the revised manuscript.

Action:

1. We have updated **Supplementary Table 6** to include enzyme loading concentrations, now

presented in both U/mL and mg/mL. These concentrations correspond to three crucial phases: pre-simulation optimization, post-first round simulation optimization, and post-second round simulation optimization. Additionally, we have provided footnotes explicitly correlating these sets of enzyme loading concentrations to **Fig. 4b**, **Fig. 4c**, and **Fig. 4d**, respectively.

2. Furthermore, to ensure comprehensive access to this critical information across all figures, we have consolidated the enzyme loading concentrations for each figure on the final two pages of this point-by-point response. Additionally, we have prepared this data in an Excel file, which can be made available to readers if needed.

The *in silico* model is an exciting and useful body of work. Why did the authors choose to show only two rounds of optimization? For instance, in Supplementary Figure 7, the model predicts that the optimal levels of TAL and RPE are low (analogous to PKL). It would follow that running additional rounds to determine if a higher levels of TAL, RPE, and PKL are optimal? Since the authors highlight in the introduction the engineering flexibility of *In vitro* biotransformation (*ivBT*), I suggest they demonstrate that by testing more than three enzyme loadings (1 U/mL, 5 U/mL, after optimization; this would also provide more data points to assess the predictive capacity of the model.

Response:

We highly appreciate your insightful suggestion. In response to this, we conducted an additional round of *in silico* optimization using our kinetic model. Our renewed efforts involved refitting the model with three distinct sets of data: the original 1-fold enzyme concentrations and the 5-fold enzyme concentrations, as well as the enzyme concentrations post-initial optimization (which correspond to Fig. 4a, 4b, and 5a in our previous manuscript, or **Fig. 4a**, **4b**, and **4c** in the revised manuscript). This refitting led to an improved model (termed Model 2), enabling subsequent simulation-based optimization of enzyme concentrations – what we refer to as "the second round of simulation optimization" in the revised manuscript. Subsequent experimental validation of the newly optimized enzyme concentrations suggested by Model 2 demonstrated a remarkable alignment between the experimental and simulation results (**Fig. 4d** in the revised manuscript), showing higher initial reaction rate and PHB titer than those of Model 1. Notably, the total enzyme concentrations remained consistent with those after the initial simulation optimization (**Supplementary Table 6**). This is a direct outcome of your invaluable suggestion, contributing significantly to achieving improved results.

Based on the results of the second round of simulation optimization (**Supplementary Fig. 10** in the revised manuscript), we observed that the enzymes PKL, TAL, and PGM were predicted to be rate-limiting, while RPE was no longer a rate-limiting enzyme. Following your suggestion, we explored the predictive capacity of the model by conducting further simulation optimization of PKL, TAL, and PGM within a broader V_{max} range (1 - 20 mM/min). As illustrated in the **Point-by-point Response Fig. 1** below, V_{max} values exceeding 5 mM/min for TAL and PGM do not yield a significantly improved initial PHB production rate. While for PKL, a higher optimal V_{max} value is suggested when expanding the scanning range from the original 1 – 5 mM/min to the broader range of 1 – 20 mM/min. Acknowledging these findings, we have maintained the methodology of the Parameter Scan tasks as detailed in the manuscript. This decision aligns with our primary objective of reducing

enzyme loading concentrations. Notably, PKL at 5 U/mL already demonstrated satisfied experimental results in terms of both reaction rate and final PHB yield (as shown in **Fig. 4b - 4d**, **Fig. 5a**, and **Fig. 5b**). Therefore, additional enhancement of PKL concentration is deemed unnecessary.

Point-by-point Response Fig. 1 | Simulation optimization of PKL, TAL and PGM within a larger range of V_{\max} . V_{\max} of 5 mM/min, highlighted by a blue circle, corresponds to the recommended optimal V_{\max} within the manuscript's specified range of 1 – 5 mM/min.

Action:

- To reflect these developments, we have updated **Fig. 4** in the manuscript to show our latest results after the second round of optimization. This revised figure provides a comparative analysis, illustrating the simulation and experimental results for all four sets of data (1-fold enzyme concentrations, 5-fold enzyme concentrations, after the first round of simulation optimization, and after the second round of simulation optimization). (**Fig. 4d** is displayed on the page 15 of this point-by-point response).
- We have added a paragraph to the end of the “Computational modeling” subsection in our Methods section, describing our recent endeavors in the second round of model fitting:

“The second round of model fitting followed the same procedures as the first round of model fitting, except that three sets of previously obtained experimental data (as displayed in Supplementary Fig. 3b, 3d, and Supplementary Fig. 9) were used. K_m , k_{eq} , C_{bind} , and n values were adjusted to fit the model to experimental data, resulting in Model 2.”
- We have renamed the "System optimization" subsection in the Results section of the revised manuscript as “*In silico* optimization of enzyme concentrations” and consolidated all *in silico* optimization results into this subsection. Our recent results regarding the second round of *in silico* optimization using the newly derived Model 2 is also included in this subsection (page 13 in the clean version):

“Recognizing the inherent complexity of our ivSEB and the challenges in developing an accurate model reflecting real-world performance, a second round of model fitting was undertaken to align simulation data with the aforementioned three sets of experimental data. During this process, adjustments were made to the k_{eq} value of α GP and the C_{bind} value of PhaC, leading to the refined Model 2 (kinetic parameters detailed in **Supplementary Table 5**). Notably, Model 2 demonstrated enhanced efficacy in simulating the data compared with

Model 1 (**Fig. 4a-4c**), particularly for data obtained after the first round of *in silico* optimization (**Fig. 4c** and **Supplementary Fig. 9**). Subsequently, a second round of *in silico* optimization using Model 2 was conducted (**Supplementary Fig. 10** and **Supplementary Table 6**). Compared with the optimized enzyme concentrations proposed by Model 1, the optimized set predicted by Model 2 mainly differed by an increase in the concentrations of α GP and a decrease in the concentrations of PhaB and 6PGDH in terms of mg/mL, while the total enzyme concentration showed minimal divergence (**Supplementary Table 6**). Guided by the outcomes of this second-round of *in silico* optimization, the experimental ivSEB achieved an accelerated initial reaction rate, closely aligning with the stimulation outcomes (**Fig. 4d**). The reaction exhibited rapid kinetics in the first four hours, followed by a deceleration, reaching equilibrium at approximately 8 h, resulting in the production of 74.9 ± 0.5 mM (7.8 ± 0.1 g/L) PHB from the consumption of 59.7 ± 1.6 mM maltodextrin at a production rate of 9.4 mM/h (approximately 1.0 g/L/h) (**Fig. 4d**). All of these values surpassed those obtained in the trial using the first-round optimized enzyme concentrations. The resulting molar yield of 125.5% was marginally lower than the first optimized trial (127.5%).”

4. Additionally, **Supplementary Fig. 10** has been added to present the outcomes of the second round of simulation optimization using Model 2. The initial results from the first round of simulation optimization using the previous model have been relocated to **Supplementary Fig. 8** for comprehensive documentation. **Supplementary Table 5** has also been updated to display all kinetic parameters pertinent to constructing the models before fitting ("Model 0"), after the first round of fitting ("Model 1"), and after the second round of fitting ("Model 2").

I found that it was important to understand that Figure 4b and Figure 5a have similar rates and titers of PHB production (though with Fig 5a utilizing reduced enzyme loading). To enhance the readers' ability to interpret the results and compare that Figure 4b and Figure 5a visually, I suggest that the flow of the paper will not be altered if Fig 4 and 5 are combined into a single figure but ultimately trust the authors intuition on this point.

Response:

Thank you for the valuable suggestion concerning Fig. 4 and Fig. 5a. In response, we have merged the outcomes detailing the conversion of 100 mM maltodextrin to PHB, achieved with varying enzyme loading concentrations, into a consolidated **Fig. 4** in our revised manuscript.

Action:

To maintain the manuscript's flow and ensure clarity, we have relocated the original Fig. 4 and Fig. 5a from the previous manuscript to the Supplementary Information, now designated as **Supplementary Fig. 3** and **Supplementary Fig. 9**, respectively. Despite this repositioning, these figures remain separate to avoid disrupting the paper's coherence.

In the main text of our revised manuscript, we have introduced an updated **Fig. 4**, which incorporates the latest findings derived from a second round of model fitting, followed by experimental validation of simulation results, that we recently conducted. This revised figure encompasses all experimental and simulation outcomes related to the one-pot, one-step PHB production from 100 mM IA-treated

maltodextrin, pre- and post-simulation optimization.

For a comprehensive overview of alterations made to Figures and Tables, two detailed tables summarizing these changes has been included at the end of this point-by-point response.

Is there a reason that the authors do not provide supplemental data (i.e. time course plots of maltodextrin consumed and PHB produced) for the data presented in Figure 5b and the claims in the text “Furthermore, to investigate the scale-up potential of our ivSEB, the starch concentration was doubled to 200 mM in a 0.5 L reactor with 0.4 L reaction solution. Accordingly, the concentrations of all enzymes (including the auxiliary enzymes), cofactors, and phosphate ions were doubled, leading to the production of 202.4 ± 2.1 mM (21.1 ± 0.2 g/L) PHB at 36 h with a slightly decreased molar yield (102.6%).” These plots would be useful to the reader.

Response:

Thank you for this valuable insight. Initially, our intention with Fig. 5b was to depict the peak PHB titer during the reaction period, inadvertently overlooking the necessity for a detailed time course plot. In response to this constructive feedback and in alignment with Reviewer #1's suggestions regarding the in-depth investigation of the 4GT and PPGK reaction step, we have addressed this by creating a new **Supplementary Fig. 13**. This figure now encompasses a comprehensive time course plot, detailing maltodextrin consumption and PHB production, thereby addressing this gap in our previous manuscript. We have also summarized the entire one-pot two-step reaction process of converting 100 mM maltodextrin into PHB in **Fig. 5a** of our revised manuscript.

Regarding the experiment utilizing 200 mM substrate, our previous data collection was constrained to specific time points (36 h and 48 h), limiting our ability to construct a comprehensive time course plot. In response to your suggestion, we recognize the importance of offering a more comprehensive view of the experiment utilizing 200 mM substrate. Therefore, we have conducted a revised experiment with enhanced sampling frequency. This approach has enabled us to construct a more detailed time course plot, which is now presented in the revised manuscript as **Fig. 5c**.

It should be noted that, despite the similarity in the final PHB titer produced from 200 mM maltodextrin between the previous 0.4-L reaction (202.4 mM) and the current trial at a milliliter scale (208.3 mM), certain variations in enzyme and the cofactor (TPP) concentrations were introduced due to our recent efforts to optimize the reaction conditions guided by you and Reviewer #1. Because of these changes, we find it logically inconsistent to keep the previous results of the 0.4-L reaction in the revised manuscript. Unfortunately, due to time constraints for manuscript revision, repeating the 0.4-L reaction at the newly optimized enzyme concentrations was not feasible. Consequently, we have chosen to omit this earlier claim in our revised manuscript. However, it is noteworthy that, one of the advantages of *in vitro* synthetic enzymatic biosystems (ivSEBs) is their linear scale-up (Dudley *et al.*, *Biotechnol J*, 2015, 10, 69-82), which has been confirmed by our previous literature for inositol production (You *et al.*, *Biotechnol Bioeng*, 2017, 114(8), 1855-1864) and our previous 0.4-L reaction for PHB production. We believe that our ivSEB with the current optimized condition could also demonstrate similar efficiency when applied to enlarged reaction volumes.

Action:

In response to your request, we have incorporated the necessary data in the revised manuscript as follows:

1. **Supplementary Fig. 13** now portrays the outcomes post the addition of different concentrations of 4GT and PPGK at 8 h, using 100 mM IA-treated maltodextrin as the substrate.

(**Supplementary Fig. 13** is displayed on the page 19 of this point-by-point response)

2. **Fig. 5** illustrates the one-pot, two-step production of PHB aiming at the complete conversion of maltodextrin, displaying a detailed time course of maltodextrin consumption and PHB production across both reaction steps.

(**Fig. 5** is displayed on the page 20 of this point-by-point response)

Table 1 is very informative in putting these results in context. Thank you for providing such clear comparisons that highlight this excellent work.

Response:

Thank you for your positive feedback on Table 1 in our manuscript. It is encouraging to know that the table effectively emphasized the key aspects of our work. Your support motivates us to continue our efforts.

Additional revisions:

In addition to addressing reviewers' comments, several amendments have been made to enhance our manuscript:

1. Revision of the Manuscript's Title

To adhere to the word limit, we have revised the title of the manuscript to “*In vitro* biotransformation of starch to acetyl-CoA-derived chemicals: An ATP-free biosystem for poly-3-hydroxybutyrate production”.

2. Update of Author Affiliations

We have revised the author affiliations by extending affiliation 2 to all authors previously associated with affiliation 1.

3. Update of Results in Abstract, Introduction, and Table 1

We conducted a second round of model fitting and optimized the loading concentrations of 4GT and PPGK, resulting in a slight improvement in the PHB yield, titer, and production rate compared to the data presented in our initial manuscript submission. These updated results have been incorporated into the Abstract, the concluding paragraph of the Introduction, and Table 1.

4. Revision of the Stoichiometric Equations

We have revised the stoichiometric equations in the Results and Discussion. The adjustments are as follows:

The modification involves relocating the numeral "4", initially a subscript for $(C_4H_6O_2)$ representing the monomer of PHB, to serve as the coefficient of $(C_4H_6O_2)$. This adjustment was made to accurately convey that each 3 glucose equivalents of substrate yield 4 monomers of PHB. The change was made to prevent any potential misunderstanding that the PHB formula is $(C_4H_6O_2)_4$.

5. Revision of Sample Legends for Fig. 3c

In order to enhance conciseness, we have streamlined the sample legends for **Fig. 3c**. The initial designations, "XPK pathway with only the essential enzymes" and "FPK pathway with only the essential enzymes," have been succinctly modified to "XPK essential" and "FPK essential," respectively. These labels now serve as brief identifiers for the systems, and we have provided a clear definition for each in the figure legend of **Fig. 3c**, along with a detailed explanation in the Results section, specifying the enzymes omitted compared to the intact system.

6. Methodology Correction

A correction was made regarding the substrate concentration in **Fig. 3c** methodology, rectifying it from 10 mM to 10 g/L (approx. 55.6 mM).

7. PHB Molecular Weight Recalculation

Because we have adjusted the experimental conditions, a re-examination of the molecular weight of PHB was conducted. Consequently, revisions were made in the main text and **Supplementary Table 7**. The updated data had no impact on the manuscript's conclusions.

8. Update of Acknowledgements

The "Acknowledgements" section has been revised to include a new funding source, the National Key R&D Program of China (2023YFA0914000), which supported our most recent research endeavors.

9. Update of References

The reference lists in the main text and in the Supplementary Information have been updated.

These amendments have been carefully implemented to ensure accuracy and completeness throughout the manuscript.

All modifications made to Tables and Figures are summarized in the two tables below for easy reference.

A Summary of Revisions of Tables

Table no.	Title	Changes made
Table 1	A comparison of different ivSEBs for PHB production	Updated with our latest experimental results.
Supp. Table 1	Equilibrium constant (k_{eq}) and Gibbs free energy change ($\Delta_r G^\circ$) values of each enzymatic reaction in the designed ivSEB	Corrected the source of PhaC. Added information of 4GT and PPGK.
Supp. Table 2	Equilibrium constant (k_{eq}) and Gibbs free energy change ($\Delta_r G^\circ$) values of each enzymatic reaction in the designed ivSEB	/
Supp. Table 3	Kinetic functions for COPASI modeling	Corrected the kinetic of FBP function by removing a factor of K_B .
Supp. Table 4	Stoichiometric coefficients of enzymatic reactions of the designed ivSEB	/
Supp. Table 5	Kinetic parameters for COPASI modeling	Updated with our latest results.
Supp. Table 6	Comparison of enzyme loading amounts prior and after simulation optimization	Added enzyme loading concentrations after the second round of simulation optimization. Showed enzyme loading concentrations in both U/mL and mg/mL.
Supp. Table 7	Molecular weights of PHB samples	Updated the results.
Supp. Table 8	Primers used for plasmid construction	/

A Summary of Revisions of Figures

Previous Figure no.	New Figure No.	Changes made	Title
Fig. 1	Fig. 1	/	Schematic of the in vitro synthetic enzymatic pathway for the production of PHB from starch.
Fig. 2	Fig. 2	/	Schematic of the in vitro synthetic enzymatic pathways for the stoichiometric production of PHB from starch.
Fig. 3	Fig. 3	/	Proof-of-concept production of PHB from debranched maltodextrin by the designed ivSEB.
/	Fig. 4	Added a new figure.	One-pot, one-step production of PHB from 100 mM debranched maltodextrin by the designed ivSEB.
/	Fig. 5	Added a new figure.	One-pot, two-step production of PHB aiming at achieving complete maltodextrin utilization.
Supp. Fig. 1	Supp. Fig. 1	/	SDS-PAGE analysis of purified enzymes used in the designed ivSEB.
Supp. Fig. 2	Supp. Fig. 2	/	Gas chromatographic profiles of methyl esters from PHB standards.
Supp. Fig. 3 ----- Fig. 4	Supp. Fig. 3	Combined the figures. Added a photo as Supp. Fig. 3e.	Production of PHB from 100 mM maltodextrin by the designed ivSEB. (with gas chromatographic results). Supp. Fig. 3e, Photos showing the reaction samples of Supplementary Fig. 3d.
Supp. Fig. 4	Supp. Fig. 4	/	Effect of NADP ⁺ input concentrations on the production of PHB.
/	Supp. Fig. 5	Added a new figure.	Effect of CoA input concentrations on the production of PHB.
/	Supp. Fig. 6	Added a new figure.	Effect of TPP input concentrations on the production of PHB.
Supp. Fig. 5	Supp. Fig. 7	Re-assigned the figure number.	Comparisons of results predicted by the COPASI kinetic models with the experimental data before and after the first round of model fitting.
Supp. Fig. 6 ----- Supp. Fig. 7	Supp. Fig. 8	Combined.	Simulation optimization of enzyme concentrations using Model 2.
Fig. 5a	Supp. Fig. 9	Moved to Supplementary Information.	A comparison of simulation results with the experimental data at optimized enzyme concentrations predicted by the first round of simulation optimization using Model 1.
/	Supp. Fig. 10	Added a new figure.	Simulation optimization of enzyme concentrations using Model 2.
/	Supp. Fig. 11	Added a new figure.	Production of PHB from edible crude starch.
/	Supp. Fig. 12	Added a new figure.	Parameter scan of V_{\max} of PhaC using Model 2 for the consumption of 100 mM substrate.
Fig. 5b	Supp. Fig. 13	Moved to Supplementary Information, and updated.	Experimental optimization of the loading amounts of 4GT and PPGK.
/	Supp. Fig. 14	Added a new figure.	Production of PHB from 200 mM glucose equivalent of maltodextrin without doubling the enzyme concentrations

A Summary of Enzyme Concentrations Used in the Main Text

	Fig. 3c, Fig. 4a		Fig. 4b		Fig. 4c		Fig. 4d		Fig. 5a		Fig. 5b		Fig. 5c	
	U/mL	mg/mL	U/mL	mg/mL	U/mL	mg/mL	U/mL	mg/mL	U/mL	mg/mL	U/mL	mg/mL	U/mL	mg/mL
αGP	1	1	5	5	2	2	3	3	3	3	3	3	6	6
PGM	1	0.022	5	0.11	3	0.07	5	0.11	5	0.11	5	0.11	10	0.22
PGI	1	0.001	5	0.006	4	0.005	3	0.003	3	0.003	3	0.003	6	0.006
PKL (XPK activity)	1	0.086	5	0.43	5	0.43	5	0.43	5	0.43	5	0.43	10	0.86
PTA	1	0.049	5	0.24	4	0.2	3	0.15	3	0.15	3	0.15	6	0.3
PhaA	0.00078	1	0.0039	5	0.00156	2	0.00156	2	0.00156	2	0.00156	2	0.00312	4
PhaB	1	0.485	5	2.43	5	2.43	4	1.94	4	1.94	4	1.94	8	3.88
PhaC	1	0.118	5	0.59	8	0.94	8	0.94	20	2.36	40	4.71	40	4.71
TAL	1	0.017	5	0.08	5	0.08	5	0.08	5	0.08	5	0.08	10	0.16
TK	1	0.02	5	0.1	4	0.08	3	0.06	3	0.06	3	0.06	6	0.12
RPI	1	0.001	5	0.006	3	0.004	1	0.001	1	0.001	1	0.001	2	0.002
RPE	1	0.001	5	0.007	5	0.007	4	0.005	4	0.005	4	0.005	8	0.01
TIM	1	0.006	5	0.03	2	0.01	1	0.006	1	0.006	1	0.006	2	0.012
ALD	1	0.002	5	0.01	2	0.004	2	0.004	2	0.004	2	0.004	4	0.008
FBP	1	0.041	5	0.21	3	0.12	5	0.21	5	0.21	5	0.21	10	0.42
G6PDH	1	0.105	5	0.53	1	0.11	1	0.11	1	0.11	1	0.11	2	0.22
6PGDH	1	0.339	5	1.69	3	1.01	1	0.34	1	0.34	1	0.34	2	0.68
4GT									0.2	0.67	0.2	0.67	0.4	1.33
PPGK									2	0.02	2	0.02	4	0.04

A Summary of Enzyme Concentrations Used in the Supporting Information

	Fig.S3a, b, Fig. S7a, c		Fig.S3c, d, e, Fig. S4-6, Fig. S7b, d		Fig. S9		Fig. S11b, c, Fig. S14		Fig. S13	
	U/mL	mg/mL	U/mL	mg/mL	U/mL	mg/mL	U/mL	mg/mL	U/mL	mg/mL
αGP	1	1	5	5	2	2	3	3	3	3
PGM	1	0.022	5	0.11	3	0.07	5	0.11	5	0.11
PGI	1	0.001	5	0.006	4	0.005	3	0.003	3	0.003
PKL (XPK activity)	1	0.086	5	0.43	5	0.43	5	0.43	5	0.43
PTA	1	0.049	5	0.24	4	0.2	3	0.15	3	0.15
PhaA	0.00078	1	0.0039	5	0.00156	2	0.00156	2	0.00156	2
PhaB	1	0.485	5	2.43	5	2.43	4	1.94	4	1.94
PhaC	1	0.118	5	0.59	8	0.94	8	0.94	20	2.36
TAL	1	0.017	5	0.08	5	0.08	5	0.08	5	0.08
TK	1	0.02	5	0.1	4	0.08	3	0.06	3	0.06
RPI	1	0.001	5	0.006	3	0.004	1	0.001	1	0.001
RPE	1	0.001	5	0.007	5	0.007	4	0.005	4	0.005
TIM	1	0.006	5	0.03	2	0.01	1	0.006	1	0.006
ALD	1	0.002	5	0.01	2	0.004	2	0.004	2	0.004
FBP	1	0.041	5	0.21	3	0.12	5	0.21	5	0.21
G6PDH	1	0.105	5	0.53	1	0.11	1	0.11	1	0.11
6PGDH	1	0.339	5	1.69	3	1.01	1	0.34	1	0.34
4GT (1-fold)									0.1	0.33
PPGK (1-fold)									1	0.01
4GT (2-fold)									0.2	0.67
PPGK (2-fold)									2	0.02

REVIEWERS' COMMENTS

Reviewer #1 (Remarks to the Author):

This reviewer is impressed with the additional work and results reported in response to the initial review. I am satisfied that nearly all of my comments have been adequately addressed. However, for the sake of the journal's and the author's scientific reputations, I still suggest that "starch-derived maltodextrin" or the equivalent be substituted for "starch" in both the title and abstract as well as at appropriate points in the manuscript. In my opinion, this does not detract from the value of the accomplishments and rather indicates that the authors and the journal are responsible contributors to the rapidly growing synthetic biology catalog of achievement.

Reviewer #3 (Remarks to the Author):

The authors successfully addressed all my comments.

I will note that the revised Figure 5 caused me a bit of confusion in panel B since "fed-batch addition of 200 mM maltodextrin" suggested to me a starting concentration of 200 mM maltodextrin which is not consistent with the author's methods or measured amount (they started with 100 mM maltodextrin).

For clarity, the authors could optionally try relabelling the title of 5b with "Optimized enzyme concentrations, fed-batch addition of 100 + 100 mM maltodextrin" and/or change the arrow label from "Add more maltodextrin and PhaC" to "Add 100 mM maltodextrin and 20U/mL PhaC"

Importantly, the figure legend is accurate and clear.

Reply to Reviewers' comments point-by-point:

Reviewer #1 (Remarks to the Author):

This reviewer is impressed with the additional work and results reported in response to the initial review. I am satisfied that nearly all of my comments have been adequately addressed. However, for the sake of the journal's and the author's scientific reputations, I still suggest that "starch-derived maltodextrin" or the equivalent be substituted for "starch" in both the title and abstract as well as at appropriate points in the manuscript. In my opinion, this does not detract from the value of the accomplishments and rather indicates that the authors and the journal are responsible contributors to the rapidly growing synthetic biology catalog of achievement.

Response and Actions Taken:

We are grateful for your positive evaluation of our additional work and results, and we sincerely value your insightful suggestion. In response to your recommendation, we have made the following modifications to the manuscript:

1. Title

- Original: " In vitro biotransformation of starch to acetyl-CoA-derived chemicals: An ATP-free biosystem for poly-3-hydroxybutyrate production"
- Revised: "ATP-free in vitro biotransformation of starch-derived maltodextrin into poly-3-hydroxybutyrate via acetyl-CoA"

2. Abstract

- Revised: "This study presents the design of an ATP-free ivSEB for one-pot PHB biosynthesis via acetyl-CoA utilizing starch-derived maltodextrin as the sole substrate." (page 2)

3. Introduction (the final paragraph)

- Revised: " In this study, we designed and constructed an ATP-free ivSEB containing 17 enzymes for the biosynthesis of PHB from maltodextrin (a derivative of starch) through acetyl-CoA." (page 5)
- Revised: "...our ivSEB produced 74.9 mM (7.8 g/L) PHB at a production rate of 9.4 mM/h (1.0 g/L/h), with a near-theoretical molar yield of 125.5% based on maltodextrin consumption." (page 5)

4. Results

- The title of the first subsection has been revised to: "Pathway design for the in vitro PHB production from **maltodextrin**" (page 5)
- Revised: "The pathway utilized by our designed ivSEB for the ATP-free production of PHB from **maltodextrin** is shown in Fig. 1. Enzymatic reactions of this pathway are described as follows: (1) **maltodextrin** is phosphorylated to glucose 1-phosphate (G1P) by α -glucan phosphorylase (α GP) ..." (page 5)
- Revised: "(5) to completely use the glucose units in **maltodextrin**, E4P and G3P are respectively recycled to F6P and Xu5P by ..." (page 6)
- Revised: "A stoichiometric coefficient of 8 for XPK in the XPK pathway (Fig. 2a) suggests for every 3 glucose equivalents of **maltodextrin** consumed by α GP, there will be ..." (page 7)
- Revised: "Therefore, the overall stoichiometric equation of our designed pathway can be written as ..., in which $C_6H_{10}O_5$ is the glucose unit of **maltodextrin**, and $C_4H_6O_2$ is the monomer unit of PHB. This equation suggests our designed ivSEB ideally consumes every 3 glucose equivalents of **maltodextrin** for the generation of 4 monomer equivalents of PHB, thus the theoretical yield of PHB from **maltodextrin** is ..." (page 8)
- Revised: "... the consumption of every 2 glucose equivalents of **maltodextrin** results in the production of 1 monomer equivalent of PHB, ..." (page 9)
- Revised: "Compared with previously reported PHB-producing ivSEBs, our system not only provided a straightforward operational process for producing PHB from **starch-derived maltodextrin**, but also achieved..." (page 16)

5. Discussion

- Revised: "Our system offers multiple advantages, including ATP independence, self-driven and self-regulating capabilities (because starch/**maltodextrin** is the sole substrate, and there is no need for coenzyme regulation modules), ..." (page 19)

6. Methods

- Revised: "..., and the auxiliary enzymes for the complete utilization of **maltodextrin** (4GT36, engineered PPGK mutant 4-1) ..." (page 21)
- Revised: "Species types were set as "fixed" for **maltodextrin** and PHB, and "reactions" for the rest of contents in the system." (page 23)
- Revised: "..., an event was set so that the simulation reaction ceased when the transient concentration of **maltodextrin** in the system was no more than 40% of the initial **maltodextrin** concentration." (pages 24)
- Revised: "Same as in the experiment, initial concentrations of **maltodextrin**, P_i , CoA, and $NADP^+$ in the model were set as ..." (page 24)

7. Table

We have changed the word “starch” into “starch-derived maltodextrin” in Table 1.

8. Figures

We have substituted the word “starch” with “maltodextrin” in Fig.1, Fig.2a and 2b, Fig.3a and 3b as well as their legends.

Fig. 1 | Schematic of the in vitro synthetic enzymatic pathway for the production of PHB from maltodextrin.

Fig. 2 | Schematic of the in vitro synthetic enzymatic pathways for the stoichiometric production of PHB from maltodextrin.

Fig. 3 | Proof-of-concept production of PHB from debranched maltodextrin by the designed ivSEB. a, Schematic of the XPK essential pathway (i.e. without TK, TIM, ALD, FBP, TAL, RPI, and PGI), in which the consumption of 2 glucose equivalents of maltodextrin results in the generation of 1 monomer equivalent of PHB. b, Schematic of the FPK essential pathway (i.e. without TK, TIM, ALD, FBP, TAL, RPI, and RPE), in which the consumption of 5 glucose equivalents of maltodextrin results in the generation of 2 monomer equivalents of PHB.

We trust these changes address your concerns, and we appreciate your guidance in maintaining the scientific accuracy and clarity of our work.

Reviewer #3 (Remarks to the Author):

The authors successfully addressed all my comments.

I will note that the revised Figure 5 caused me a bit of confusion in panel B since "fed-batch addition of 200 mM maltodextrin" suggested to me a starting concentration of 200 mM maltodextrin which is not consistent with the author's methods or measured amount (they started with 100 mM maltodextrin).

For clarity, the authors could optionally try relabelling the title of 5b with "Optimized enzyme concentrations, fed-batch addition of 100 + 100 mM maltodextrin" and/or change the arrow label from "Add more maltodextrin and PhaC" to "Add 100 mM maltodextrin and 20U/mL PhaC"

Importantly, the figure legend is accurate and clear.

Response and Actions Taken:

We sincerely appreciate your positive feedback on our manuscript and the constructive nature of your comments. Your insights have been instrumental in refining the clarity of our presentation. In response to your suggestion, we have relabeled the title of Fig. 5b:

- Original: "Optimized enzyme concentrations, fed-batch addition of 200 mM maltodextrin"
- Revised: "Optimized enzyme concentrations, fed-batch addition of 100 + 100 mM maltodextrin"

We have also changed the arrow labels:

- Original labels: "Add more maltodextrin and PhaC" and "Add 4GT, PPGK"
- Revised labels: "Add extra maltodextrin and PhaC" and "Add 4GT, PPGK and polyP", the latter label applies to Fig.5a and 5c as well

Fig. 5 | One-pot, two-step production of PHB aiming at achieving complete maltodextrin utilization. **a**, Production of PHB from 100 mM glucose equivalent of maltodextrin. Initial reaction conditions were the same as those of Fig. 4d (refer to Methods for details), except that PhaC was added at 20 U/mL as suggested by Model 2 (see Supplementary Fig. 12). At 8 h, 4GT, PPGK, and polyphosphate (**polyP**) in the form of sodium hexametaphosphate were added to the system, reaching final concentrations of 0.2 U/mL (approximately 0.67 mg/mL), 2 U/mL (approximately 0.02 mg/mL), and 20 mM, respectively. **b**, Production of PHB from 200 mM glucose equivalent of maltodextrin using a fed-batch substrate addition strategy. Initial reaction conditions aligned with those of Fig. 5a. At 4 h, another 100 mM maltodextrin and 20 U/mL PhaC was added. At 12 h, 4GT, PPGK, and **polyP** were added to the system at the same concentrations as in Fig. 5a. **c**, Production of PHB from 200 mM glucose equivalent of maltodextrin with a single substrate addition. Enzymes, NADP⁺, CoA, TPP, and phosphate ion concentrations were doubled compared to those in Fig. 5a and 5b. At 8 h, 4GT, PPGK, and **polyP** were added to the system at doubled concentrations as in Fig. 5a and 5b. Reactions were performed in triplicate (n = 3 biologically independent samples) and data are presented as mean values ± SD. Source data are provided as a Source Data file.

The arrow labels without concentrations maintain a clear and uncluttered figure. Comprehensive details on the additions are available in the figure legend, ensuring transparency and understanding for the readers.

We trust the modification addresses your concerns and contributes to the overall clarity of the figure. Your thoughtful feedback has been invaluable in refining the accuracy and presentation of our work. Thank you for your time and expertise in reviewing our research.

Additional revisions:

In addition to addressing reviewers' comments, we have also made the following modification:

Revision of PHB titers in g/L

For the conversion of PHB titers from mM monomer equivalents to g/L, our previous manuscript wrongly used the molecular weight of 3HB (around 104) instead of that of a PHB monomer (around 86), therefore, all the PHB titers in g/L in the revised main manuscript were re-calculated. We have also revised the related information in Table 1.

The updated data had no impact on the manuscript's conclusions.